# SUMOylation controls the binding of hexokinase 2 to mitochondria and protects against prostate cancer tumorigenesis

Xun Shangguan [1,3], Jianli He[1,2,3], Zehua Ma[1], Weiwei zhang [1], Yiyi Ji[1], Kai Shen[1], Zhiying Yue[1], Wenyu Li [1], Zhixiang Xin[1], Quan Zheng[2], Ying Cao[2], Jiahua Pan[1], Baijun Dong[1], Jinke Cheng [1,2✉], Qi Wang [1✉] & Wei Xue [1✉]

Human hexokinase 2 is an essential regulator of glycolysis that couples metabolic and proliferative activities in cancer cells. The binding of hexokinase 2 to the outer membrane of mitochondria is critical for its oncogenic activity. However, the regulation of hexokinase 2 binding to mitochondria remains unclear. Here, we report that SUMOylation regulates the binding of hexokinase 2 to mitochondria. We find that hexokinase 2 can be SUMOylated at K315 and K492. SUMO-specific protease SENP1 mediates the de-SUMOylation of hexokinase 2. SUMO-defective hexokinase 2 preferably binds to mitochondria and enhances both glucose consumption and lactate production and decreases mitochondrial respiration in parallel. This metabolic reprogramming supports prostate cancer cell proliferation and protects cells from chemotherapy-induced cell apoptosis. Moreover, we demonstrate an inverse relationship between SENP1-hexokinase 2 axis and chemotherapy response in prostate cancer samples. Our data provide evidence for a previously uncovered posttranslational modification of hexokinase 2 in cancer cells, suggesting a potentially actionable strategy for preventing chemotherapy resistance in prostate cancer.

[1] State Key Laboratory of Oncogenes and Related Genes, Department of Urology, Renji Hospital, School of Medicine, Shanghai Jiao Tong University, Shanghai, China. [2] Shanghai Key Laboratory for Tumor Microenvironment and Inflammation, Department of Biochemistry and Molecular Cell Biology, Shanghai Jiao Tong University School of Medicine, Shanghai, China. [3] These authors contributed equally: Xun Shangguan, Jianli He. ✉email: jkcheng@shsmu.edu.cn; wqi@sjtu.edu.cn; xuewei@renji.com

Most cancer cells rely on aerobic glycolysis rather than oxidative phosphorylation, a phenomenon termed the Warburg effect. They demand high levels of nutrients, especially glucose, to fulfill nutrient requirements for rapid growth and proliferation. Hexokinases (HK), the first rate-limiting enzymes of glycolysis, catalyze the conversion of glucose to glucose-6-phosphate. This irreversible enzymatic reaction is of fundamental importance not only because it traps glucose inside cells but also because its product glucose-6-phosphate is at the convergence point of glycolysis, the pentose phosphate pathway, the hexosamine pathway, and glycogen synthesis. In mammalian cells, there are five known hexokinase isoforms encoded by separate genes[1]. HK1 is constitutively expressed in multiple tissues; HK2 is expressed in embryonic tissue and aggressive tumors, such as lung cancer, hepatocyte cell cancer, breast cancer, and prostate cancer[2–4]; HK3 and HK5 (known as HKDC1) are poorly characterized[5]; HK4, also known as glucokinase, is located primarily in the liver and the endocrine pancreas. Among these hexokinases, HK2 is predominant in malignant or rapidly pro-liferating tumors rather than most normal adult tissues. Of significance, deleting *HK2* inhibits tumor progression with no sign of adverse physiological effects[2,3]. These properties warrant the consideration of HK2 as an attractive target for antitumor therapy.

Beyond the well-studied function of HK2 in glucose metabolism, accumulating evidence has revealed that HK2 also plays a critical role in cell death and apoptosis. HK2 can bind to mTOR complex 1 and facilitate autophagy during glucose starvation[6]. In cardiomyocytes, HK2 is required for AKT-mediated mitochondrial protection against the opening of the mitochondrial permeability transition pore. In particular, the oncogenic potential of HK2 is controlled by its cellular localization, a process that relies on the binding of HK2 to VDAC1 on outer membrane mitochondria. Mitochondria-associated HK2 competes with proa-poptotic proteins, such as Bax, to prevent the release of cytochrome *c*, which may trigger the intrinsic pathway of cell apoptosis[7]. Mitochondrial binding deficient mutant (MTD) HK2, while still retaining hexokinase activity, failed to promote cell proliferation[3]. Therefore, binding to mitochondria is required for HK2 oncogenic function. However, the regulation of HK2 subcellular localization remains unclear.

SUMOylation, one of the key posttranslational modifications, is a critical event in the dynamic regulation of multiple cellular processes. Similar to ubiquitination, SUMOylation is mediated by E1, E2, and E3. In mammalian cells, there is only one E2 enzyme, UBC9, which can conjugate SUMO to the target protein. The reversible progression of De-SUMOylation is mediated by the SENP family. SUMOylation is known to target its canonical consensus motif $\psi$KXE ($\psi$ is a large hydrophobic amino acid, $X$ any amino acid, and $K$ is the site of SUMO conjugation). Extensive studies have linked SUMOylation to diverse target protein regulation, such as stability, structure, function, activity, location, and interaction with other proteins. For example, previous studies showed that PTEN can be SUMOylated at K254 and K266 and then recruited to the plasma membrane to suppress the PI3K-AKT pathway[8]. Another group reported that PTEN with SUMOylation at K254 was released from the nucleus upon DNA damage stress[9]. In another study, SUMOylation enhanced the interaction of CREB with PP2A and regulated brown adipocyte differentiation[10].

In this study, we demonstrate that HK2 is the direct target of SUMOylation and that SENP1 mediates the de-SUMOylation of HK2. SUMO-defective HK2 preferably binds to mitochondria and enhances glycolysis. This metabolic reprogramming supports prostate cancer cell proliferation and protects cells from chemotherapy-induced cell apoptosis. Our data also demonstrate

an inverse relationship between the SENP1-HK2 axis and che-motherapy response in human prostate cancer samples.

## Results

**HK2 can be SUMOylated in prostate cancer cells.** In a previous study, stable isotope labeling with amino acid, a quantitative proteomic technique, revealed that endogenous HK2 in PC3 cells is a putative target for protein SUMOylation[11]. To determine whether HK2 is subjected to SUMOylation, we first performed immunoprecipitation and western blotting with HK2 and UBC9, the sole SUMOylation-conjugating enzyme in mammalian cells. We exogenously expressed HA-tagged *HK2* and Flag-tagged *UBC9* in human embryonic kidney 293T cells. Figure 1a con-firmed the association of HK2 and UBC9. Next, we expressed HA-tagged HK2, UBC9, and Flag-tagged SUMO1 or SUMO2 in 293T cells and then immunoprecipitated cell lysates with anti-Flag followed by western blotting with anti-HA. The results showed that HK2 is mostly conjugated by SUMO1 in a UBC9-dependent manner (Fig. 1b). To verify HK2 SUMOylation in prostate cancer cells, we used PC3, which contains high levels of endogenous HK2[4], to perform a coimmunoprecipitation assay. Figure 1c shows an enriched SUMOylated HK2 band with a molecular weight of 122 kDa (the expected normal size of HK2 is 102 kDa) in PC3. These results indicated that HK2 covalently conjugated with one molecule of SUMO1 in PC3 cells.

**SUMO1 is conjugated to both Lys 315 and Lys 492 on the HK2 protein.** To determine the candidate SUMOylation site of HK2, we adopted the GPS-SUMO (http://sumosp.biocuckoo.org/online.php) and JASSA analysis tools and found two lysine resi-dues, K315 and K492, with high scores (Supplementary Fig. 1a)[12,13]. Interestingly, these two lysine residues are highly evolutionally conserved in various species in HK2 (Fig. 1d) but not HK1 (Supplementary Fig. 1c). This analysis indicated that SUMOylation may only occur on HK2. Indeed, in Fig. 1f, western blotting showed no SUMO band for HK1 in the immunopreci-pitation with SUMO1. To further confirm this hypothesis, we immunoprecipitated cell lysate with an anti-HK1 antibody and performed western blotting with the SUMO antibody. Consistent with the above results, no SUMO band of HK1 was observed in PC3 cells (Supplementary Fig. 1c). By using site-directed muta-genesis, we created K315R mutant, K492R mutant, and K315R/K492R double-mutant (DKR) constructs of rat HK2 (Supple-mentary Fig. 2a) and stably expressed these mutation forms in PC3 and LNCaP cells; endogenous *HK2* was knocked down with human shRNA so that each mutant was predominantly expressed (Supplementary Fig. 2b, c). We co-expressed His-tagged SUMO1 and UBC9 and HA-tagged HK2 (wild-type or different mutants) and detected SUMOylated HK2 in 293T cells. Figure 1e shows that the HK2 SUMO band dramatically diminished in HK2 K315/492R double-mutant samples. Because SUMOylation is critically dependent on the glutamic acid at +2 (E) of the acceptor lysine (K), we then generated E317A/E494A (DEA) mutant and found that the SUMOylated bands of DEA were notably reduced compared to those of wild-type HK2 (Supplementary Fig. 2d). This result strongly indicated that HK2 can be conjugated with one molecule of SUMO1 at K315R and K492R. Notably, we did not observe a shift in HK2 from 122 to 142 kDa, which pre-sumably represents HK2 conjugated with two molecules of SUMO, suggesting that K315 and K492 are not simultaneously SUMOylated in cells. To determine whether mutating both lysines could affect the structure of HK2, we generated human HK2 3D protein models SWISS-MODEL (https://swissmodel.expasy.org/)[14]. Supplementary Fig. 2e shows no difference

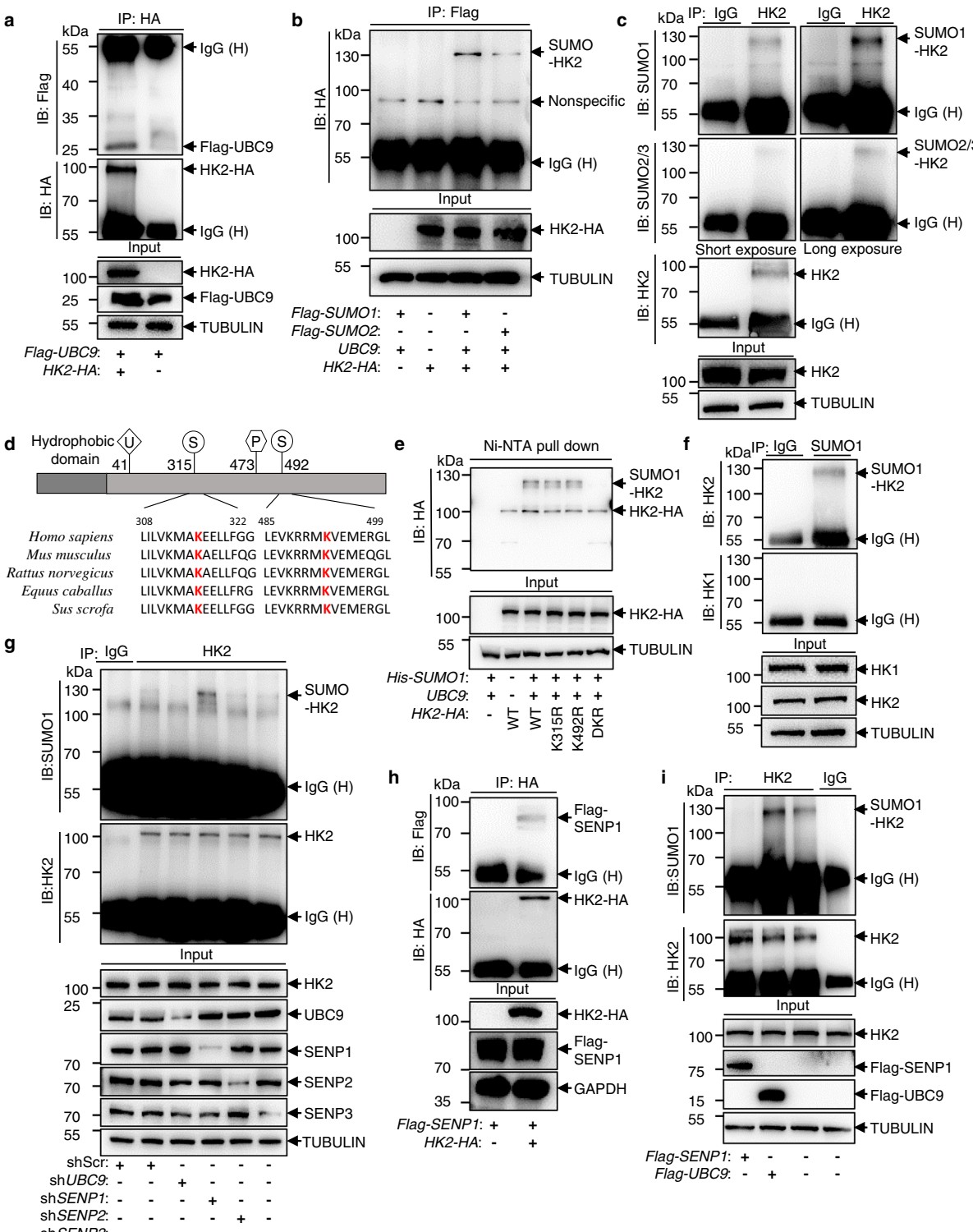

between the structure of wild-type and K315R/K492R mutant HK2.

Because SENPs can de-SUMOylate target proteins and play a major role in the SUMOylation process, we next investigated which SENPs mediate HK2 SUMOylation. In PC3 cells with stable knockdown of *UBC9, SENP1, SENP2,* or *SENP3*, we observed that the SUMO HK2 band decreased in *UBC9* knockdown cells but increased in *SENP1* knockdown cells compared to that in control cells (Fig. 1g). We also confirmed the interaction between HK2 and SENP1 by co-expressing these

two plasmids and performing immunoprecipitation and western blotting (Fig. 1h). Consistently, we exogenously expressed Flag-tagged SENP1 or UBC9 in PC3 cells and showed that SUMOylated HK2 increased when *UBC9* was overexpressed and decreased when *SENP1* was overexpressed (Fig. 1i). These results suggest that HK2 can be SUMOylated on K315 and K492 and that SENP1 is the major de-SUMOylating enzyme for HK2.

**SUMOylation of HK2 does not affect hexokinase activity, protein stability, or phosphorylation.** To determine the

**Fig. 1 HK2 can be SUMOylated at Lys 315 and Lys 492, and de-SUMOylated by SENP1. a** Coimmunoprecipitation of HK2 and UBC9 protein in HEK293T cells transfected with *Flag-Ubc9* and *HK2-HA* plasmids. **b** SUMO1 and SUMO2 ligation with HK2 protein in HEK293T cells transfected with *HK2-HA*, *UBC9*, and *Flag-SUMO1/Flag-SUMO2*. **c** Endogenous HK2 SUMOylation was detected by immunoprecipitation with IgG or anti-HK2 antibody and then western blotting with anti-SUMO1 or anti-SUMO2/3 antibodies. **d** Sequence alignment of HK2 homologs in various species. The potential SUMOylation site is denoted in red. **e** 293T cells were transfected with HA-tag HK2 wild-type (WT), *HK2*-K315R, *HK2*-K492R, or *HK2*-DKR with or without *His-SUMO1* and *UBC9*. Cell lysates were prepared for precipitation with $Ni^{2+}$-NTA resin, followed by western blotting with indicated antibodies. **f** A reciprocal immunoprecipitation assay was performed with IgG or SUMO1 antibody for the lysates of PC3 cells, and then western blotting with HK1 and HK2 antibodies. **g** *SENP1* depletion enhanced HK2 SUMOylation. *UBC9*, *SENP1*, *SENP2*, or *SENP3* was stably knocked down by shRNA in PC3 cells. Cell lysates were prepared for precipitation with HK2 antibody-conjugated protein A/G agarose beads, and SUMOylation was detected by SUMO1 antibody. **h** Coimmunoprecipitation of HK2 and SENP1 protein in HEK293T cells transfected with *Flag-SENP1* and *HK2-HA* plasmids. **i** Overexpression *SENP1* decreased while *UBC9* increased SUMOylation of HK2. Flag-UBC9 or Flag-SENP1 was transfected into PC3 cells, followed by detection of HK2 SUMOylated bands with SUMO1 antibody. Source data are provided as a Source Data file.

functional consequences of HK2 SUMOylation, we stably expressed wild-type or different mutant forms of HK2 in PC3 and LNCaP cells, replacing endogenous HK2 by knockdown with shRNA. First, we measured hexokinase activity in PC3 and LNCaP cells. Consistent with a previous study, knockdown of *HK2* or S155/603A-kinase-dead mutant HK2 led to a significant reduction in hexokinase activity compared to that in the control[3]. However, K315R and K492R mutation HK2 did not affect kinase activity (Supplementary Fig. 3a). To assess the protein stability, we incubated PC3 cells in 20 μM cycloheximide (CHX), a protein synthesis inhibitor, for 24 h and monitored HK2 protein expression. Neither the K315R nor the K492R mutation in HK2 altered protein stability (Supplementary Fig. 3b). Stable knockdown of *UBC9* or *SENP1* did not affect HK2 protein stability (Supplementary Fig. 3c). We also detected HK2 ubiquitination in PC3 and showed that K315R and K492R site mutations had no effect on HK2 ubiquitination (Supplementary Fig. 3d). Because a previous study showed that AKT can phosphorylate HK2 at T473[15], we then investigated whether these two posttranslational modifications can affect each other. PC3 cells expressing the *HK2* SUMO site mutation did not alter the phosphorylation of HK2 (Supplementary Fig. 3e). We also created the *HK2* T473 mutation and showed that phosphorylation did not alter the SUMOylation of HK2 (Supplementary Fig. 3f).

**SUMOylation controls the binding of HK2 to the mitochondria**. In the protein structure, the first helix on the N-terminus of HK2 is considered to be the mitochondrial-binding peptide. We generated the 3D-structure of the human HK2 protein at http://www.rcsb.org (PDB 2nzt). Based on this model, K315, one of the SUMO sites of HK2, is located near the N-terminus of HK2 (Supplementary Fig. 2f). Therefore, we speculated that SUMOylation might influence HK2 subcellular localization. We immunoprecipitated endogenous HK2 and immunoblotted VDAC1 in PC3 cells. Knockdown of *UBC9* strongly increased the binding of HK2 to VDAC1, while knockdown of *SENP1* decreased the binding compared to that in the control (Fig. 2b). As expected, the mitochondrial binding deficient mutant HK2 did not bind to VDAC1. However, the K315/492R mutation, but not the single mutation, increased the binding of HK2 to VDAC1 (Fig. 2a). By co-staining anti-HA HK2 immunofluorescence (green) and MitoTracker™ Red CMXRos (red), we observed the subcellular localization of HK2 with a super-resolution microscope. While wild-type HK2 was located in both the mitochondrion-free cytosol and mitochondria, K315/492R mutant HK2 staining largely overlapped with mitochondrial staining (Fig. 2c). Moreover, knockdown of *UBC9* increased the overlap, while knockdown of *SENP1* decreased the overlap (Fig. 2e). Figure 2d and f shows the quantification analysis by ImageJ. Given that HK2 binds to VDAC1 on the outer mitochondrial membrane, we fractionated PC3 cells to obtain both mitochondrial and mitochondrion-free

cytosolic fractions, followed by western blotting to detect HK2 in each fraction. Knockdown of *UBC9* resulted in a marked increase in the level of HK2 in the mitochondrial fraction and a decrease in HK2 in the mitochondrion-free cytosolic fraction; in contrast, knockdown of *SENP1* increased the HK2 level in the mitochondrion-free cytosolic fraction and decreased the HK2 level in the mitochondrial fraction. Consistently, MTD HK2 was released from mitochondria to the mitochondrion-free cytosol, while K315/492R-mutated HK2 accumulated mostly in the mitochondrial fraction (Fig. 2h, i). Moreover, we next immunoprecipitated the mitochondrial or mitochondrion-free cytosolic fraction of the cell lysates with anti-HK2 and subjected them to western blotting with anti-SUMO1. The results showed that more SUMOylated HK2 was expressed in the mitochondrion-free cytosolic fraction than in the mitochondrial fraction (Fig. 2g). In summary, these data strongly suggest that SUMO-defective HK2 more robustly binds to VDAC1 on the outer mitochondrial membrane.

**SUMO-defective HK2 increases prostate cancer cell glycolysis**. Because HK is the first rate-limiting enzyme for cell glycolysis and plays a critical role in glucose metabolism, we investigated the role of HK2 SUMOylation in glucose uptake and subsequent metabolism. Consistent with a previous report, *HK2* knockdown or mitochondrial binding deficient mutation dramatically decreased glucose uptake and lactate production compared to those in the control. However, K315/492R mutation led to increased glucose uptake and lactate production in different prostate cancer cells (Fig. 3a, b and Supplementary Fig. 4a, b), demonstrating that SUMO-defective HK2 enhances glycolytic flux in prostate cancer cells. To further quantify glycolytic function and mitochondrial respiration, we used the Seahorse XF bioenergetic system to analyze HK2 cells with different mutants. The results showed that the K315/492R mutant HK2 resulted in a higher glycolytic rate and lower internal respiratory capacity compared with wild-type HK2 (Fig. 3c, d). Reactive oxygen species (ROS) are viewed as an inevitable byproduct of mitochondrial oxidative phosphorylation. We used a flow cytometer assay to quantify intracellular ROS levels by using DCFH-DA and mitochondrial ROS levels by using MitoTracker™ Red CMXRos. While MTD HK2 moderately increased intracellular ROS level, partially from mitochondria, K315/492R HK2 decreased both intracellular and mitochondrial ROS levels (Fig. 3e) Graphically account for all FACS sequential gating strategies were summarized in Supplementary Fig. 7. These data suggest that SUMO-defective HK2 promotes cell glycolysis and inhibits mitochondrial respiration.

**SUMO-defective HK2 contributes to prostate cancer cell proliferation and tumorigenesis**. In addition to its role in glucose

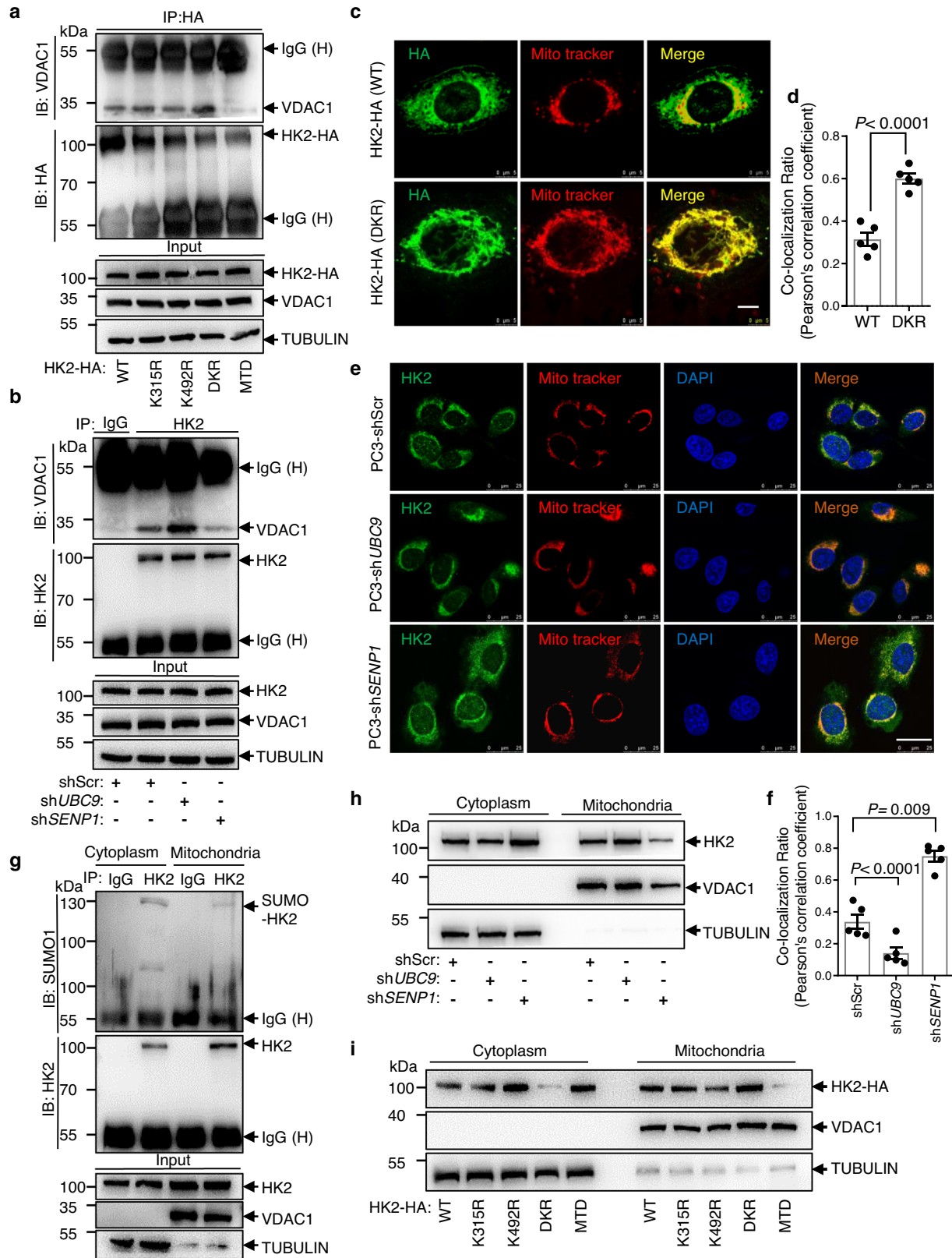

metabolism regulation, HK2 has been reported to be over-expressed in multiple aggressive tumors and plays a key role in promoting cancer cell proliferation. Previous research also shows that binding to VDAC1 on the outer mitochondrial membrane is critical for HK2 to promote tumorigenesis[3]. To explore whether SUMOylation of HK2 modulated the cancer cell aggressive phenotype, we performed proliferation curve analysis, BrdU incorporation, and anchorage-independent colony-formation assays to test cell proliferative activity and oncogenesis with different HK2 mutations. While knockdown of endogenous HK2 markedly reduced cell proliferation, ectopic expression of wild-type but not MTD HK2 rescued the effect of HK2 silencing.

**Fig. 2 SUMOylation controls HK2 binding to the mitochondria. a** SUMO-defective HK2 preferentially interacted with VDAC1. 293T cells were transfected with *HA*-tag *HK2* WT, *HK2*-K315R, *HK2*-K492R, *HK2*-DKR, or MTD (mitochondrial binding deficient). Cell lysates were precipitated with HA antibody and blotted by VDAC1 antibody. **b** SUMOylation of HK2 prevented its interaction to VDAC1. Endogenous HK2 and VDAC1 interaction was accessed by immunoprecipitation with IgG or anti-HK2 antibody and then western blotting with anti-VDAC1. **c, d** Representative fluorescent images showed colocalization of ectopic expression HK2 and mitochondria in PC3 cells. Wild-type or DKR mutant HK2 with HA tag was expressed in PC3 cells and detected by HA antibody (green), co-staining of mitochondria with MitoTracker[TM] Red CMXRos (Red). Scale bar, 5 μm. Quantification is calculated with Pearson's correlation using ImageJ. Data are presented as mean ± SEM of five biologically independent samples. Statistical significance was determined by a two-tailed Student's *t* test. **e, f** Representative fluorescent images showed colocalization of endogenous HK2 and mitochondria in PC3 cells. Scramble control or *UBC9/ SENP1* knockdown PC3 cells were stained with HK2 antibody (green) and MitoTracker[TM] Red CMXRos (Red). Scale bar, 25 μm. Quantification is calculated with Pearson's correlation using ImageJ. Statistical significance was determined by a two-tailed Student's *t* test. **g** Endogenous SUMOylation of HK2 was detected in the mitochondria and mitochondria-free cytosolic fractions in PC3 cells. Immunoprecipitation was performed with IgG or HK2 antibody, and then western blotting with SUMO1 antibody. **h** Western blotting showed endogenous HK2 expression in mitochondrial and cytoplasmic extraction from *UBC9* shRNA, *SENP1* shRNA or scramble PC3 cells. **i** PC3 cells expressing different mutant forms of HK2 were prepared for mitochondrial and cytoplasmic extraction and probed by HA and VDAC1 antibody in western blotting. Source data are provided as a Source Data file.

K315/492R mutant HK2 even accelerated cell proliferation compared with wild-type (Fig. 4a–c). To test tumorigenic potential in vivo, PC3 cells were used in subcutaneous tumorigenesis assays. Consistently, mice containing *HK2* shRNA cells had markedly smaller tumors than control groups. Wild-type HK2 restored tumor growth, and K315/492R mutant HK2 promoted tumorigenesis more than wild-type HK2 (Fig. 4d–f). All these data demonstrate that SUMO-defective HK2 might contribute to prostate cancer cell proliferation and oncogenesis.

**SUMO-defective HK2 desensitizes chemotherapy response in prostate cancer cells.** A considerable fraction of prostate cancer patients eventually progresses to androgen-deprivation therapy, a condition known as castration-resistant prostate cancer. Docetaxel, a taxane antimitotic agent, is currently used as a first-line chemotherapeutic treatment for these patients. However, only 50% of the patient's response to docetaxel treatment[16–18]. Given that HK2 inhibition sensitizes cancer cells to multiple chemotherapy agents[19–22], we speculated that HK2 SUMOylation may play roles in docetaxel treatment response. Indeed, docetaxel treatment increased cellular ROS levels and mitochondrial respiration in PC3 cells (Supplementary Fig. 5a, b), and SUMO-defective HK2 decreased ROS and mitochondrial OCR (Fig. 3d, e). It is conceivable that SUMO-defective HK2 may protect cells from docetaxel treatment. To test this possibility, we treated PC3 cells with 20 nM docetaxel for 48 h and collected the surviving cells. We then immunoprecipitated cell lysates with anti-HK2 antibody and western blotting with the anti-SUMO1 antibody. Docetaxel treatment significantly decreased HK2 SUMOylation in PC3 cells (Fig. 5a). To test whether docetaxel treatment affected HK2 subcellular localization, we co-stained endogenous HK2 (green) and mitochondria (red) in surviving cells. Compared with DMSO, cells treated with docetaxel largely overlapped with the mitochondria (Fig. 5b). These data suggested that in the surviving cells treated with docetaxel, HK2 was prone to de-SUMOylation and binding to mitochondria. Because SENP1 is the major enzyme for HK2 de-SUMOylation, we then addressed the question of whether the SENP1-HK2 axis is affected by docetaxel treatment. A previous study confirmed that SENP1 is the direct target of HIF-1α, which plays a significant role in cancer cell drug resistance and hypoxia[23]. In NCBI's Gene Expression Omnibus database (GSE83654)[24], three prostate cancer cell lines were treated with docetaxel and collected for microarray. By analyzing these data, we found that the HIF-1 pathway was significantly activated in a time-dependent manner (Fig. 5c). Moreover, gene set enrichment analysis also revealed a large fraction of HIF pathway genes that were differentially expressed between docetaxel-resistant and docetaxel-sensitive prostate cancer cells (Fig. 5d), indicating that HIF signaling was active in docetaxel-resistant cells. By collecting

surviving PC3 cells after docetaxel treatment, we confirmed the increased HIF-1α and SENP1 mRNA and protein levels by qPCR and western blotting, respectively (Fig. 5e, f). Similar results were obtained when we replaced PC3 with prostate-specific antigen (PSA)-positive cell lines LNCaP and 22Rv1. Docetaxel could upregulate HIF-1α and SENP1 with or without androgen-deprived treatment (Supplementary Fig. 5c). These data suggested that docetaxel treatment may activate the HIF-1 pathway and then increase *SENP1* expression.

To further confirm the role of the SENP1-HK2 axis in docetaxel treatment, we detected the viability of cells expressing different mutant forms of HK2 under docetaxel treatment. In PC3 cells, knockdown of *HK2* sensitized cells to docetaxel treatment, and SUMO-defective HK2 increased cell viability even more than wild-type HK2 in both a time- and dose-dependent manner (Fig. 5g and Supplementary Fig. 5d). These results also confirmed in LNCaP cells, with or without charcoal-stripped serum (mimicking androgen deprivation). SUMO-defective HK2 protects cells from docetaxel treatment, regardless of with or without androgen-deprived treatment (Fig. 5h). By Annexin V/PI flow cytometry and cleaved-caspase 3 detection, we found that SUMO-defective HK2 protected cells from docetaxel-induced cell apoptosis and death (Supplementary Fig. 5e, f). To assess the in vivo effect of HK2 on docetaxel treatment in prostate cancer, mice-bearing PC3-derived prostate xenografts were treated with three injections of docetaxel (10 mg/kg i.p.), and tumor volume was examined every week. As shown in Supplementary Fig. 5g, PC3-derived xenografts with SUMO-defective HK2 conferred a survival advantage compared with HK2 wild-type, with or without docetaxel treatment (Supplementary Fig. 5g). To determine that the advantage of these xenografts is related to cell apoptosis, we detected the expression of cleaved-caspase 3 through immunohistochemistry in the xenograft tumors, and the results were consistent with previous in vitro findings, suggesting that tumor with SUMO-defective HK2 was inversely associated with cell apoptosis and death. To further validate these results in PSA positive cell line, we developed 22Rv1-derived xenograft with docetaxel treatments, in intact or castrated hosts. SUMO-defective HK2 still offered a survival advantage in 22Rv1 cells and eliminated cell apoptosis, regardless of whether mice had been castrated (Supplementary Fig. 5i, j). Taken together, these in vitro and in vivo data indicate that in addition to promoting cell proliferation, SENP1-HK2 may protect prostate cancer cells from docetaxel-induced cell apoptosis and death.

**Upregulation of the SENP1-HK2 axis is associated with poor outcomes and worse chemotherapy response in prostate cancer patients.** We sought to assess the potential effect of HK2 SUMOylation on prostate cancer patient prognosis. Since there is

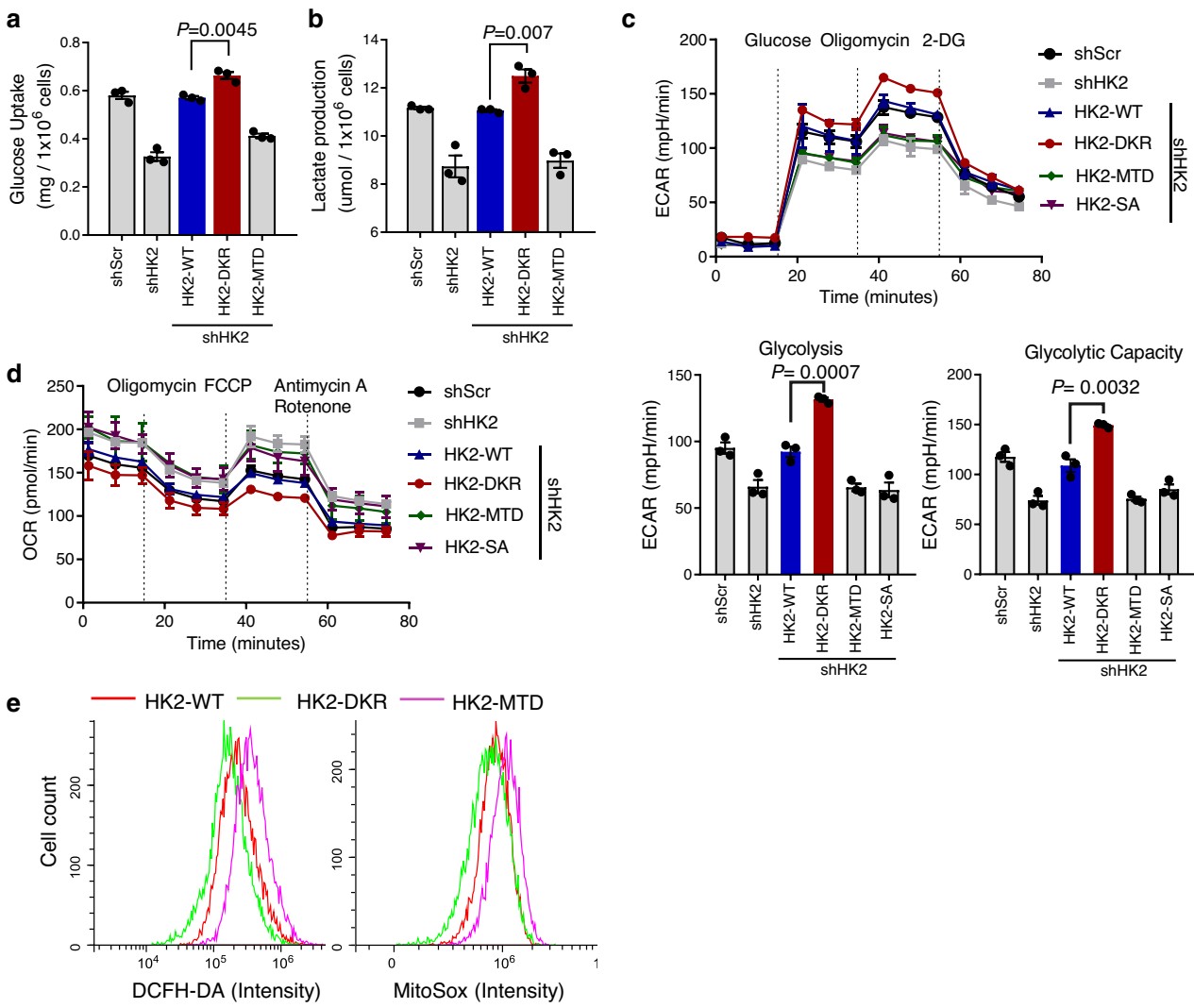

**Fig. 3 SUMO-defective HK2 increases prostate cancer cell glycolysis and decreases mitochondrial respiration and ROS. a** Glucose consumption was measured in PC3 cells with different mutant forms HK2. Endogenous *HK2* was knockdown by shRNA and replaced by different mutant forms with HA tag in PC3 cells. Data are presented as mean ± SEM of three biologically independent samples. Statistical significance was determined by a two-tailed Student's *t* test. **b** Lactate production was measured in PC3 cells with different mutant forms HK2. Data are presented as mean ± SEM of three biologically independent samples. Statistical significance was determined by a two-tailed Student's *t* test. **c** PC3 cells as described were measured by the mitochondrial stress kit to determine extracellular acidification rate (ECAR). Data are presented as mean ± SEM of three biologically independent samples. Statistical significance was determined by a two-tailed Student's *t* test. **d** PC3 cells as described were measured by the mitochondrial stress kit to determine oxygen consumption rate (OCR). **e** Intracellular and mitochondrial levels of reactive oxygen species (ROS) were measured in indicated PC3 cells by flow cytometry. Left: PC3 with the different mutant form of HK2 was loaded with ROS probe DCFH-DA, and DCF fluorescent intensity was measured by flow cytometry. Right: Cells labeled with MitoTracker™ Red CMXRos were measured by flow cytometry to see the mitochondrial level of ROS. Experiments were performed at least twice in triplicates. Source data are provided as a Source Data file.

no available antibody specific for SUMOylated HK2, the next best thing is to assess the correlation of HK2 and SENP1 levels in patient samples with patient outcomes. We performed immunohistochemical staining in a tissue microarray of prostate cancer samples collected as previously described. These patients underwent radical prostatectomy without any drug treatment. We stained the tissue microarray with HK2 and SENP1 antibodies and scored the staining on a scale of 0–3 based on the percentage of immunoreactive tumor cells and the staining intensity. Scores of 0 and 1 were marked as low levels, while scores of 2 and 3 were marked as high. Representative images of cancer tissue immunostaining for HK2 and SENP1 are shown in Fig. 6a. We found that patients with high expression of both HK2 and SENP1 had significantly higher Gleason scores and shorter biochemical progression-free survival times than patients with low expression

(Fig. 6b, c). These results suggested that high expression of both HK2 and SENP1 indicates a poor prognosis in prostate cancer patients. We further investigated data from a large cohort of prostate adenocarcinoma patients from The Cancer Genome Atlas. Consistently, the data demonstrated a significant positive correlation between Gleason score and high levels of both *HK2* and *SENP1*. We analyzed the disease-free survival time in these patients, which were also categorized by *HK2* and *SENP1* levels. Of clinical significance, the prognosis was the worst in patients with both high *HK2* and high *SENP1* (Supplementary Fig. 6a, b). Furthermore, high *HK2* and *SENP1* expression also indicated the worst prognosis in a subgroup with a Gleason score = 7 (Supplementary Fig. 6c). In summary, high expression of both *HK2* and *SENP1* might predict poor outcomes in prostate cancer patients.

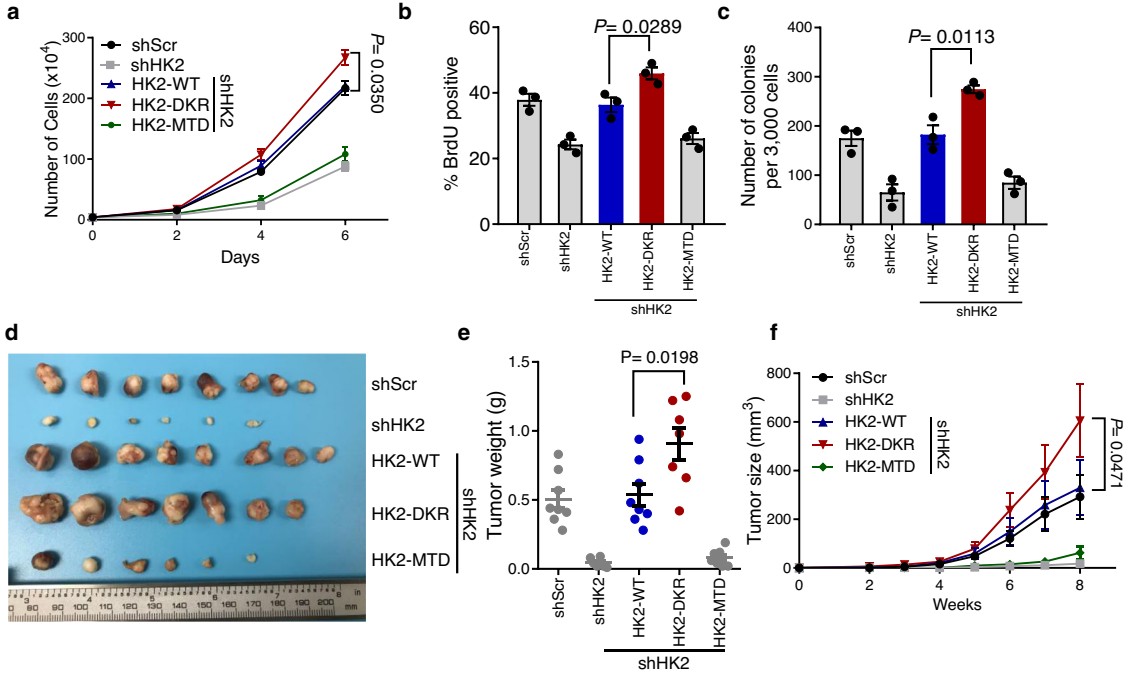

**Fig. 4 SUMO-defective HK2 contributed to prostate cancer cell proliferation and tumorigenesis. a** Cell proliferation curve of PC3 cells was determined by regular cell counting. PC3 stably expressed different mutant forms of HK2 as indicated, instead of endogenous HK2. Data are presented as mean ± SEM of three biologically independent samples. Statistical significance was determined by a two-tailed Student's $t$ test. **b** BrdU incorporation analysis in PC3 cells. Bar graph shows the proportion of BrdU-positive cells among total cells. Data are presented as mean ± SEM of three biologically independent samples. Statistical significance was determined by a two-tailed Student's $t$ test. **c** Anchorage-independent growth of PC3 cells was accessed by soft agar. Data are presented as mean ± SEM of three biologically independent samples. Statistical significance was determined by a two-tailed Student's $t$ test. **d** Subcutaneous tumors formed in nude mice by PC3 cells as described at 8 weeks ($n = 8$/group). **e** Tumor weight was measured at the endpoint of the study. Each bar represents the mean ± SEM. Statistical significance was determined by a two-tailed Student's $t$ test. **f** Tumor formation growth curves were recorded every week after subcutaneous injection in nude mice. Data are represented as means ± SEM. Statistical significance was determined by a two-tailed Student's $t$ test. Source data are provided as a Source Data file.

In addition, we collected prostate cancer samples from patients who received neoadjuvant therapy before radical prostatectomy. One group received neoadjuvant hormonal therapy; the other group received docetaxel-based neoadjuvant chemohormonal therapy, as described in Supplementary Table 4. We stained the tissue microarray with HK2 and SENP1 antibodies and scored the staining based on positive percentage and intensity, as described above. We analyzed the percentage of patient biochemical recurrence in 1 year and found that in the neoadjuvant hormonal therapy group, HK2 and SENP1 levels were not significantly correlated with biochemical recurrence (Supplementary Fig. 6d); however, in the neoadjuvant chemohormonal therapy group, high expression of both HK2 and SENP1 was strongly correlated with a shorter time to biochemical recurrence than low expression, suggesting that activation of the SENP1-HK2 axis is associated with disease progression (Fig. 6d). In summary, our study provides both in vitro and in vivo evidence that activation of the SENP1-HK2 axis might associate with poor docetaxel response (Fig. 6e).

## Discussion

HK2, by physically and functionally interacting with the mitochondria, couples metabolic activity and cell proliferation in cancer cells. Cellular localization is critical for HK2 oncogenic function. In this study, we aimed to explore the role of HK2 SUMOylation in the modulation of its subcellular trafficking, especially during chemotherapy in prostate cancer. We have shown for the first time that SUMO1 conjugates to HK2 at K315 and K492 and that SENP1 mediates the de-SUMOylation of HK2.

Considering that one of the SUMOylation sites, K315, is near the N-domain of HK2, the mitochondrial binding site in the 3D protein structure (Supplementary Fig. 2f), it is reasonable to suggest that SUMOylation may affect HK2 binding to mitochondria. Indeed, our data demonstrate that SUMO-defective HK2 preferentially interacts with VDAC1 and therefore binds to the outer membrane of mitochondria.

Posttranslational modification of HK2 plays an important role in the assembly of the HK2/VDAC1 complex and therefore may regulate HK2 binding to the mitochondria. For example, AKT phosphorylates HK2 at T473 to maintain the association of HK2 with the mitochondria. However, this modification did not show the distinction between HK1 and HK2[15,25,26]. HK2 has also been shown to be the ubiquitin target in prostate cancer cells through the covalent binding of the ubiquitin E3 HectH9. HectH9 promotes HK2 ubiquitination and regulates HK2 localization to the mitochondria in prostate cancer stem cells[27]. Our current study adds SUMOylation to the repertoire of HK2 posttranslational modifications. SUMO conjugates to HK2 and hinders the interaction of HK2 and VDAC1. Interestingly, immunoprecipitation and subsequent western blotting data suggest that this modification did not target HK1. Indeed, the sequence of the conserved $\psi$KXE SUMOylation sites on HK2, K315, and K492, are not conserved in HK1 (Supplementary Fig. 1b). Since HK1 functions as a housekeeping gene in normal tissue and HK2 is upregulated in most aggressive tumors, this distinction makes the SENP1-HK2 axis an excellent target for tumor therapy.

Accumulating evidence shows that prostate cancer undergoes metabolic reprogramming and that HK2 is required for this

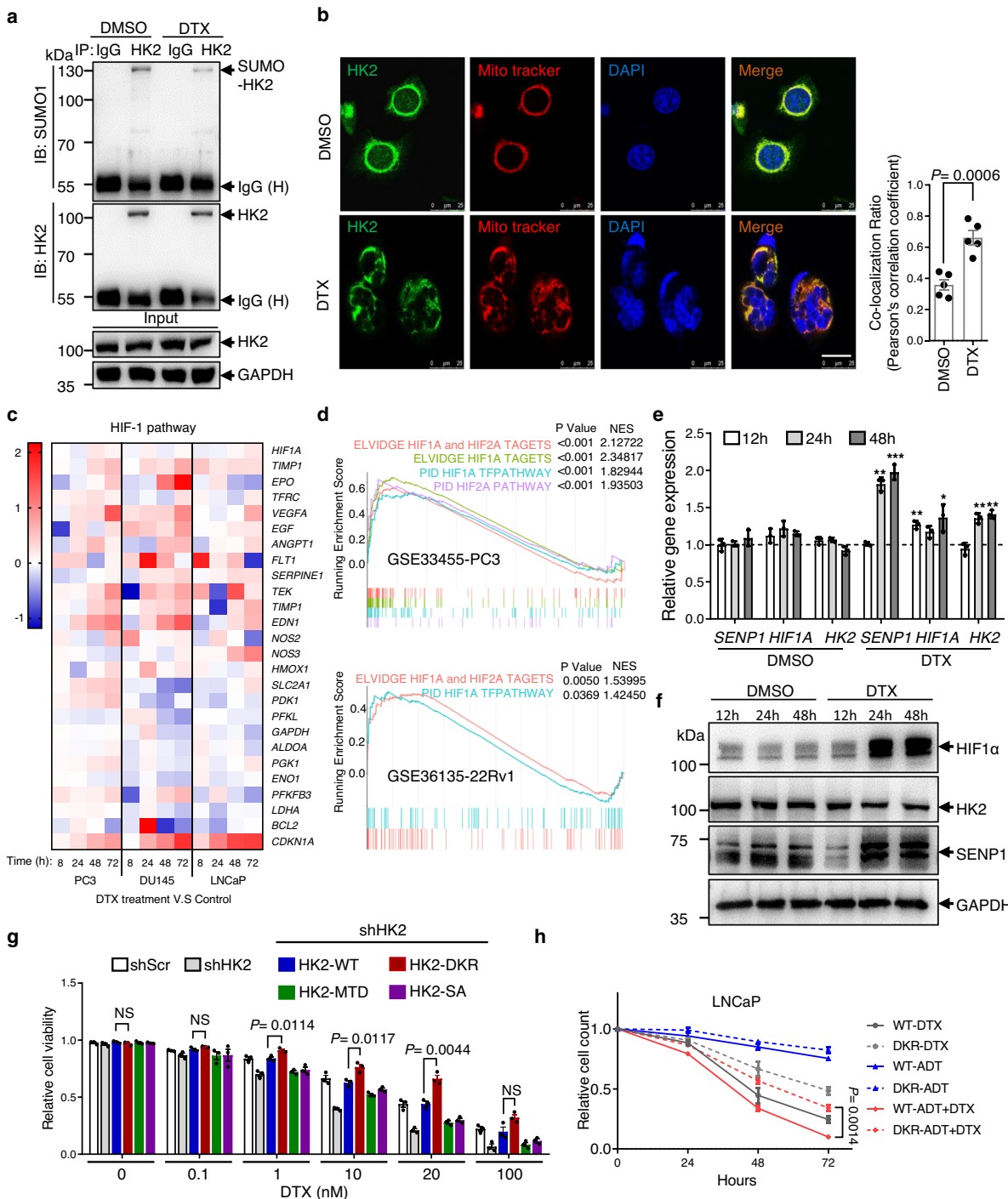

process[27–29]. HK2 levels are high in prostate cancer, especially in association with poor prognosis. In the $Pten^{PC-/-}/Tp53^{PC-/-}$ prostate cancer mouse model, *HK2* is required for tumorigenesis. Prostate-specific *PTEN* deletion activates the AKT-mTORC1 pathway and increases *HK2* mRNA translation, while *TP53* loss inhibits miR-143 biogenesis and enhances *HK2* mRNA stability[30].

In another study of a *Pten*-deficient prostate tumor mouse model, systemic deletion of HK2 inhibited tumor growth and metastasis[4]. These studies and our findings collectively illustrate that HK2 plays a critical role in cell proliferation and metabolism changes during prostate cancer initiation and progression. Moreover, our study extends the role of HK2 in docetaxel

**Fig. 5 SUMO-defective HK2 conferred to cell chemotherapy resistance. a** Docetaxel treatment decreased endogenous HK2 SUMOylation. SUMO band was detected by immunoprecipitation with IgG or anti-HK2 antibody and then western blotting with anti-SUMO1 antibodies. **b** Colocalization analysis of endogenous HK2 with MitoTracker[TM] in PC3 cells with or without docetaxel treatment. Left: Representative fluorescent images show endogenous HK2 and MitoTracker[TM] in PC3 cells. Scale bar, 25 μm. Right: Colocalization between HK2 and MitoTracker[TM] was quantified by calculating the Pearson's correlation using ImageJ. Data are presented as mean ± SEM of five independent samples. Statistical significance was determined by a two-tailed Student's *t* test. **c** Heatmap represented the effect of docetaxel treatment to LNCaP, DU145, and PC3 cells on HIF-1 pathway genes at indicated time points. Data from the NCBI's Gene Expression Omnibus database (GEO GSE83654). **d** Gene set enrichment plots of HIF pathway associated genes with docetaxel resistance in PC3 cells (upper panel, GSE33455, *n* = 3 independent experiments), or 22Rv1 (lower panel, GSE36135, *n* = 3 independent experiments). *P* value is determined by GSEA software. **e** qPCR showed relative RNA level of *HIF1A*, *HK2*, and *SENP1* in PC3 cells at the indicated time points with or without docetaxel treatment. Data are presented as mean ± SEM of three independent samples. Statistical significance was determined by a two-tailed Student's *t* test. *P* value: *P < 0.05, **P < 0.01, ***P < 0.001. **f** Western blotting analysis of HIF-1α, HK2, and SENP1 expression at the indicated time points in PC3 cells with or without docetaxel treatment. **g** CCK-8 assay of cell viability in different concentrations of docetaxel treatment for 48 h. PC3 stably expressed a different mutant form of HK2 as indicated. Data are presented as mean ± SEM of three independent samples. Statistical significance was determined by a two-tailed Student's *t* test. NS: not significant. **h** LNCaP cells cultured in charcoal-stripped medium (ADT) with or without 5 nM docetaxel were observed for cell viability at the indicated time points. Data are presented as mean ± SEM of three independent samples. Statistical significance was determined by a two-tailed Student's *t* test. Source data are provided as a Source Data file.

response in prostate cancer chemotherapy. Docetaxel treatment can activate HIF-1α, and HIF-1α regulates the HK2 pathway via diverse molecular mechanisms. First, consistent with a previous study, which suggested that HIF-1α binds to the *HK2* promoter and promotes *HK2* transcription[31], our data showed that docetaxel treatment upregulates HIF-1α, concomitant with *HK2* upregulation. Second, HIF-1α can increase *SENP1* expression, which mediates the de-SUMOylation of HK2 and promotes HK2 binding to mitochondria. This subcellular trafficking of HK2, therefore, protects cells from docetaxel-induced cell apoptosis. It is worth noting that because of the high level of *SENP1* in PC3 cells, HK2 remains in de-SUMOylated status in these cells. Indeed, cells with SUMO-defective HK2, which preferentially binds to the mitochondria, display higher survival rates than cells with WT HK2 upon docetaxel treatment (Fig. 5g). In agreement with these cell experiments, the histopathological analysis showed that in prostate cancer patient samples, upregulation of both HK2 and SENP1 is associated with poor prognosis and poor response to docetaxel-based chemotherapy (Fig. 6d). Taken together, our data provide evidence of SUMO modification of HK2 and an outline of the related mechanism, shedding light on alternative strategies for prostate cancer therapy.

## Methods

**Plasmids and lentiviral production**. The plenti6-rat*HK2*-HA-WT, rat*HK2*-HA-MTD, rat*HK2*-HA-SA plasmids were kindly provided by Prof. Nissim Hay (the University of Illinois at Chicago, Chicago, IL 60607, USA). Mutations of *HK2* were obtained from PCR-directed mutagenesis using KOD-plus Kit (TOYOBO). shRNAs against target genes were generated with pLKO.1 vector (sequences are shown in Supplementary Table 1). Lentivirus was prepared using a three-plasmid packing system. Briefly, pLKO.1 or plenti6 vectors were co-transfected into 293T cells along with expression vectors containing the VSVG and Δ8.9 genes. Lentivirus was harvested at 24, 48, and 72 h after transfection, and the virus was cleansed by the 0.45-μm filter. Stable cell lines were selected out in 2 μg/mL puromycin or 10 μg/mL blasticidin S for 1 week. Cells were expanded for two passages in drug-free media and frozen for subsequent use. Early passage cells were used for every experiment.

**Cell culture**. The human prostate cancer cell lines, LNCaP, 22Rv1, PC3, and 293T human embryonic kidney cells were purchased from the Cell Bank, Shanghai Institutes for Biological Sciences, Chinese Academy of Sciences, where they were recently authenticated by short tandem repeat (STR) profiling and characterized by mycoplasma and cell vitality detection. These were cultured at 37 °C in a humidified incubator (5% CO₂) in RPMI-1640 medium or DMEM (Gibco, Carlsbad, CA, USA) supplemented with 10% fetal bovine serum (FBS), 100 U/mL penicillin, and 100 μg/mL streptomycin. All cell lines were verified negatively for mycoplasma contamination by MycoAlert[TM] Mycoplasma Detection Kit (Lonza, LT07-418). No cell lines used here appear in the database of commonly misidentified cell lines (the International Cell Line Authentication Committee).

**Western blotting**. Cells were lysed in RIPA lysis buffer and the protein concentration of the cell lysates determined by BCA Protein Assay (Thermo Scientific, 23227). Equal amounts of protein were loaded onto SDS-PAGE gel and transferred to PVDF membranes. Western blotting was performed using primary antibodies and secondary antibodies conjugated with HRP. For immunoblotting, the following antibodies were used: anti-SENP2 (Abcam, ab58418, 1:1000), anti-HIF-1α (Cell Signaling Technology, 79233, 1:1000), anti-phospho-threonine (Cell Signaling Technology, 9386, 1:1000), anti-Ubc9 (Cell Signaling Technology, 4786, 1:1000), anti-BrdU (Cell Signaling Technology, 5292, 1:1000), anti-ubiquitin (Cell Signaling Technology, 3936, 1:1000), anti-cleaved-caspase 3 (Cell Signaling Technology, 9661, 1:1000), anti-SUMO1 (Cell Signaling Technology, 4930, 1:1000), anti-SUMO2/3 (Cell Signaling Technology, 4971, 1:1000), anti-SENP1 (Cell Signaling Technology, 11929, 1:1000), anti-SENP3 (Cell Signaling Technology, 5591, 1:1000), anti-HA-Tag (Cell Signaling Technology, 3724, 1:2000), anti-Flag-Tag (Cell Signaling Technology, 14793, 1:2000), anti-hexokinase 1 (Proteintech, 19662-1-AP, 1:1000), anti-hexokinase 2 (Proteintech, 22029-1-AP, 1:1000), anti-VDAC1 (Proteintech, 10866-1-AP, 1:1000), anti-alpha tubulin (Proteintech, 66031-1-Ig, 1:5000), anti-GAPDH (Proteintech, 10494-1-AP, 1:5000).

**SUMOylation assays by Ni $^{2+}$-NTA pull down**. HK2 SUMOylation was analyzed with in vivo SUMOylation assay using Ni$^{2+}$-NTA beads in HEK-293T cells, as previously described[32,33]. Briefly, HEK-293T cells were transfected with 1 μg of each of HA-HK2 (WT or Mutants), Flag-UBC9, and His₆-SUMO1-expressing plasmids. Forty-eight hours after transfection, cells were collected and 25% of cells were lysed by M-RIPA buffer for western blotting. The remaining cells were lysed in 3 mL of His-lysis buffer. In total, 60 μL of Ni$^{2+}$-NTA-agarose beads (Qiagen) were then added to the lysates and incubated at 4 °C overnight. The beads were successively washed for 5 min in each step at room temperature with 750 μL of each of the washing buffers 1–4. After the last wash, the beads were incubated in 75 μL of elution buffer for 20 min at room temperature. The eluates were analyzed by western blotting. His-lysis buffer (0.01 M Tris/HCl, pH 8.0, 0.1 M Na₂HPO₄/NaH₂PO₄, 5 mM imidazole, 6 M guanidinium-HCl, and 10 mM β-mercaptoethanol). Washing buffer 1 (0.01 M Tris/HCl, pH 8.0, 0.1 M Na₂HPO₄/NaH₂PO₄, 6 M guanidinium-HCl, and 10 mM β-mercaptoethanol). Washing buffer 2 (0.01 M Tris/HCl, pH 8.0, 0.1 M Na₂HPO₄/NaH₂PO₄, 8 M urea, and 10 mM β-mercaptoethanol). Washing buffer 3 (0.01 M Tris/HCl, 0.1 M Na₂HPO₄/NaH₂PO₄, pH 6.3, 8 M urea, 10 mM β-mercaptoethanol, and 0.2% Triton X-100). Washing buffer 4 (0.01 M Tris/HCl, 0.1 M Na₂HPO₄/NaH₂PO₄, pH 6.3, 8 M urea, 10 mM β-mercaptoethanol, and 0.1% Triton X-100). Elution buffer (0.15 M Tris/HCl pH 6.7, 30% glycerol, 200 mM imidazole, 0.72 M β-mercaptoethanol, and 5% SDS).

**SUMOylation analysis by immunoprecipitation**. The method was described[34] with several modifications. For analysis of endogenous SUMO1-HK2 protein, PC3 cells were grown in 10-cm plates. Cells were collected in NEM-PBS and the cell pellets lysed by adding 200 μL of SUMO lysis buffer (62.5 mM Tris pH 6.8, 2% SDS) and boiling for 10 min. The samples were centrifuged for 20 min at full speed in an Eppendorf microcentrifuge. The supernatant was transferred to a new tube and either stored at −80 °C until required for further analysis or used for direct protein determination. This lysate was diluted 1/20 with M-RIPA buffer. Immunoprecipitation with 2 μL of anti-HK2 antibody was used, and immunoprecipitants were resolved by SDS-PAGE and immunoblotted with the anti-SUMO1 antibody.

**RNA isolation and real-time quantitative PCR**. The total RNA was isolated from cells by Trizol reagent (Invitrogen, CA, USA). Complementary DNA was

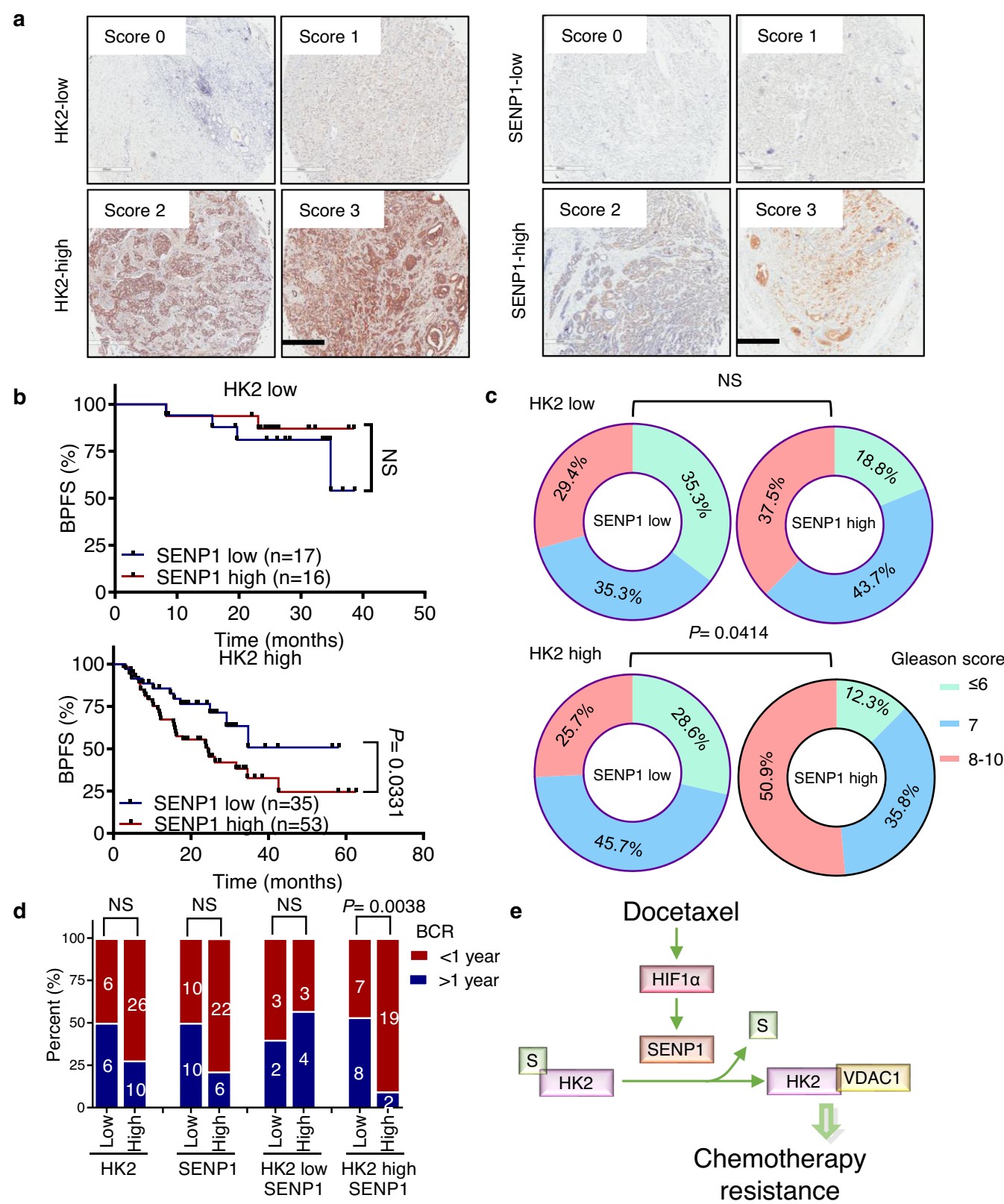

synthesized using a cDNA synthesis kit (Takara, Shiga, Japan) according to the manufacturer's instructions. Fluorescence real-time PCR was performed with SYBER Green Premix (Takara, Shiga, Japan) using the Bio-Rad CFX Manager 3.0 System (Bio-Rad, Hercules, CA, USA). PCR was carried out in triplicate and standard deviations representing experimental errors were calculated. Pairs of PCR primers used to amplify the target genes are presented in Supplementary Table 2.

**Measurements of glucose consumption and lactate production.** A total of $1 \times 10^6$ PC3 cells suspended in 3 mL of medium were seeded in 60-mm dishes, and the medium was changed after 12 h with no serum RPMI-1640. Cells were incubated for 20 and 8 h for measurement of glucose consumption and lactate production, respectively. The culture medium was then collected for measurement of the glucose and lactate concentrations. Glucose levels were determined using a glucose (GO) assay kit (Sigma, GAGO-20). Glucose consumption was defined as the

**Fig. 6 HK2 together with SENP1 upregulation associates poor outcome and worse chemotherapy response in prostate cancer patients. a** Representative images of immunohistochemistry staining of HK2 and SENP1 in Renji prostate cancer tissue microarray. Scores were calculated by intensity and percentage of stained cells. Scale bar, 600 μm. **b** Kaplan–Meier survival analysis in HK2 high subgroup of prostate cancer biochemical progression-free survival (BPFS) defined as SENP1 low or high expression, using Renji tissue microarray cohort. Statistical difference was determined by a two-sided log-rank test. NS not significant. **c** Relationship between SENP1 staining and Gleason Score in HK2 low or high subgroup, using Renji tissue microarray cohort. Gleason Score was divided in low ≤6, medium = 7, and high 8–10. The distribution of high Gleason Score increased in SENP1 and HK2 both high group. Statistical significance was measured by the chi-square test. NS not significant. **d** The percentage of biochemical recurrence <1 or ≥1 year of docetaxel-based neoadjuvant chemohormonal therapy prostate cancer patients ($n = 48$ samples), stratified by HK2 and SENP1 expression. Statistical significance was measured by the chi-square test. NS not significant. **e** Schematic diagrams illustrating the critical role of HK2 SUMOylation in prostate cancer cell docetaxel treatment. Source data are provided as a Source Data file.

difference in glucose concentration in the medium with or without cell incubation. Lactate levels were determined using a D-Lactate colorimetric assay kit (Sigma, MAK058). Cells were collected and counted, and glucose consumption and lactate production were normalized according to cell number (per $10^6$ cells).

**Measurement of intracellular ROS levels**. ROS levels were determined by using a fluorescent dye 2′,7′-dichlorofluorescin diacetate (DCFH-DA, Sigma, D6883). Briefly, cells with specified treatments were washed with PBS and incubated with 10 μM DCFH-DA at 37 °C for 30 min to load the fluorescent dye. Afterward, cells were washed twice with PBS and trypsinized for ROS detection. Fluorescence (Ex.488 nm, Em.525 nm) was monitored by FACS using a flow cytometer. CytExpert 2.0 software was used for data analysis.

**Oxygen consumption rate (OCR) and extracellular acidification rate (ECAR)**. OCR and ECAR were determined using the XF96 Extracellular Flux Analyzer (Seahorse Bioscience). Briefly, $2 \times 10^4$ PC3 cells were plated onto XF96 plates and incubated at 37 °C, 5% CO₂ overnight, pretreated with or without docetaxel (20 nM) for 48 h. Then, cells were washed with Seahorse buffer (RPMI-1640 with phenol red containing 25 mM glucose and 2 mM glutamine). Cell Mito Stress Test Kit was used to measure cellular mitochondrial function, 175 μL of Seahorse buffer plus 25 μL each of 1 μM oligomycin, 0.5 μM FCCP, and 1 μM rotenone were automatically injected to determine the OCR, according to the manufacturer's instructions. The Glycolysis Stress Test Kit was used to measure the glycolytic capacity, 25 μL each of 10 mM glucose, 1 μM oligomycin, and 100 mM 2-deoxyglucose (2-DG) were added to determine the ECAR, according to the manufacturer's instructions.

**Hexokinase activity assay**. To examine hexokinase activity a colorimetric assay was performed as per the manufacturer's instructions (Sigma, MAK091). Briefly, $1 \times 10^6$ cells were lysed in ice-cold assay buffer and then centrifuged, and 10 μL of homogenate was added to 96-well plates. Reaction mix was added to each of the wells and the product of enzyme reaction, which results in a colorimetric product proportional to the enzymatic activity. The absorbance at 450 nm was recorded by incubating the plate at 37 °C taking measurements every 5 min for 30 min.

**Cell proliferation and BrdU incorporation**. Cells ($4 \times 10^4$/dish) were plated in 6-cm dishes in triplicate and counted every 2 days for 6 days. We changed the media on the 3rd day to keep cells' continuous natural growth. For BrdU incorporation, cells were labeled with 3 μg/mL BrdU for 2 h and fixed by 70% ethanol for 30 min. Cells were stained with primary anti-BrdU monoclonal antibodies, followed by Alexa 488-conjugated secondary antibody. Then BrdU-positive cells were detected using a flow cytometer. CytExpert 2.0 software was used for data analysis.

**Soft agar colony assay**. The effect of HK2 and its mutants on anchorage-independent growth was assessed using a soft agar colony assay. Briefly, this assay was performed in six-well plates with a base of 2 mL of medium containing 10% FBS with 0.6% Bacto agar (Amresco). Stable PC3 cells were seeded in 2 mL of medium containing 10% FBS with 0.35% agar at $1 \times 10^3$ cells per well and layered onto the base, respectively. The photographs of the cells growing in the plate and of the colonies developed in soft agar were taken, and the number of colonies was scored by ImageJ V1.45 (NIH, USA).

**Cell viability assay**. PC3 cells were seeded in 96-well plates. The next day, fresh media containing Docetaxel (Selleck, S1148) (0.1, 1, 10, 20, 100 nM), or control (0.1% DMSO) were added and cells incubated for 3 days. Cell Counting Kit-8 (CCK-8) from MedChemExpress (Madison, WI, USA) was used to determine cell viability, as per the manufacturer's instructions. Cell viability was normalized against the vehicle control, and the data expressed as a percentage of control from three independent experiments done in triplicate.

**Cell death assays**. PC3 cells were seeded in six-well plates. The next day, fresh media containing docetaxel (20 nM), or control (0.1% DMSO) were added and cells

incubated for 2 days. After treatment, the cells were trypsinized, collected, and stained with annexin V-fluorescein isothiocyanate (FITC) and propidium iodide (PI) simultaneously using an Annexin V-FITC Apoptosis Detection kit (Invitrogen, V13242). The cell suspensions were analyzed with a Beckman Coulter flow cytometer to determine the percentage of apoptotic (FITC stained cells) and necrotic cells (PI-stained cells). CytExpert 2.0 software was used for data analysis.

**Immunofluorescence and confocal microscopy**. PC3 were seeded into the uncoated 35-mm dishes at a density of $1.0 \times 10^5$ cells. After 48 h, the medium was replaced with fresh media containing 100 nM MitoTracker Red CMXRos (Invitrogen, M7512) at 37 °C for 15 min. Cells were washed in cold PBS twice, fixed with 4% freshly prepared formaldehyde in PBS for 8–10 min, and then washed three times with PBS. Cells were permeabilized with 0.2% Triton-X-100)/PBS for 15 min, blocked in 5% normal goat serum for 30 min, incubated in the primary antibodies anti-HK2 (Proteintech, 22029-1-AP, dilution 1:400) or anti-HA (Cell Signaling Technology, 3724, dilution 1:500) diluted in blocking solution for 2 h, washed three times with PBS and then incubated in the second antibody (Donkey Anti-Rabbit IgG H&L, Alexa Fluor® 488, Abcam-ab150061, dilution 1:500) in blocking solution for 1 h. The cells were then washed three times with PBS. DAPI (4′,6′-diamidino-2-phenylindole, Sigma, D9542) was added for DNA staining.

Images were taken with a Zeiss LSM710 Confocal Microscope (Carl Zeiss, Jena, Germany). All confocal images were analyzed and quantified using ImageJ v. 1.45 (http://rsb.info.nih.gov/ij/).

**Isolation of the mitochondria and protein fractionation**. Mitochondria of PC3 cells were isolated by conventional differential centrifugation as described previously[35]. Briefly, PC3 cell pellets were harvested, washed once with cold PBS, and then resuspended in an isolation buffer (10 mM Tris-HCl, pH 7.5, 10 mM NaCl, 1.5 mM MgCl₂, 1 mM EDTA, 70 mM sucrose, 210 mM mannitol, and protease inhibitors. After chilling on ice for 10 min, the cell suspension was disrupted with 15 strokes in a glass homogenizer. The homogenate was centrifuged twice at 1500×g at 4 °C for 5 min to remove unbroken cells and nuclei. The supernatants were centrifuged at 15,000×g for 15 min to separate the mitochondrial fraction and mitochondrial-free cytosolic fraction. The mitochondria fractions were then pelleted at 16,000×g for 10 min.

**Subcutaneous tumor implantation**. Male BALB/c nude mice were subcutaneously inoculated with $1 \times 10^6$ of PC3 or 22Rv1 cells (with HK2 depletion or HK2 reconstitution) cells. Tumor size was measured every week by a caliper with the formula $0.52 \times L \times W^2$, where $L$ indicates length and $W$ indicates width. For tumorigenesis assay, all mice were sacrificed and tumors were harvested 8 weeks later, followed by photography and weighing. For docetaxel treatment assay, animals were grouped and drug administration commenced when tumor volume reached approximately 150 mm³. Mice were killed 3 days after the last docetaxel administration, and the tumors were collected for photography and immunohistochemistry analysis of the expression of cleaved-caspase 3. Docetaxel was administered i.p. at 10 mg/kg weekly, for three treatments. Docetaxel or vehicle control was administered in a volume of 10 μL/g body weight. All mice were maintained in the animal facility at Renji Hospital, School of Medicine, Shanghai Jiao Tong University under specific-pathogen-free (SPF) conditions. Animals were housed in groups of four to five mice per individually ventilated cage in a 12 h light/dark cycle (07:30–19:30 light. 19:30–7:30 dark), with controlled room temperature (23 ± 2 °C) and relative humidity (40–50%).

**Expression profile and gene set enrichment analysis**. Expression profiles of The Cancer Genome Atlas (TCGA) human prostate cancer data set were downloaded from The cBioPortal for Cancer Genomics (http://cbioportal.org)[36] and are summarized in Supplementary Table 3. The gene expression profile of DU145, LNCaP, and PC3 Prostate Cancer Cell Lines upon Docetaxel treatment (GSE83654)[24] were downloaded from The Gene Expression Omnibus (GEO, http://www.ncbi.nlm.nih.gov/geo/)[37]. The gene expression profile of docetaxel-sensitive or -resistant PC3 cells (GSE33455)[38] and docetaxel-sensitive or -resistant 22Rv1 cells (GSE36135)[39] were downloaded and are analyzed using gene set enrichment analysis (GSEA)[40] to

identify the ranked list of genes affected by the resistance to docetaxel. Gene sets identified as related to biological signal conduction on the MSigDB (http://software.broadinstitute.org/gsea/msigdb), which may be found on the GSEA website. The thresholds for inclusion were P less than 0.05 and q less than 0.25.

**Human prostate cancer tissue microarray and immunohistochemistry analysis.** Tissue samples and clinical parameters of 121 prostate cancer patients who underwent radical prostatectomy were collected; these are listed in Supplementary Table 4. Tissue samples from high-risk or locally advanced prostate cancer patients who received hormonal or chemohormonal therapy prior to radical prostatectomy were collected. The brief treatment and follow-up strategy were described previously[41]. The clinical information of these patients is listed in Supplementary Table 5.

Prostate tumor tissues were fixed with formalin and embedded in paraffin. A tissue core size of 1 mm containing the dominant tumor area was collected to construct a tissue microarray. For immunohistochemistry analysis, the paraffin-embedded tissue sections were deparaffinized and rehydrated. Antigen retrieval was carried out with a 10 mM citrate acid repair buffer (pH 6.0) at 95 °C for 30 min. A 3% $H_2O_2$ solution was used to block endogenous peroxidase activity. Blocking buffer (10% horse serum/TBS-T) was added to the slides and incubated for 30 min at room temperature. Slides were then incubated with indicated primary antibodies diluted in blocking buffer at 4 °C overnight. The slides were then washed in TBS-T and incubated by an anti-rabbit EnVisionTM kit (DAKO, Glostrup, Denmark) for 30 min at 37 °C. Sections were then counterstained with hematoxylin, dehydrated with ethanol, and mounted under coverslips. For SENP1 and HK2 quantification in human tissue samples, two observers independently scored the degree of immunostaining and were clinically blind. The staining extent score was on a scale of 0–3, corresponding to the percentage of immunoreactive tumor cells and the staining intensity. For cleaved-caspase 3 in mice tumors, five nonoverlapping fields in each section of mice tumor were analyzed in a blinded manner. The integral optical density of immunopositive cancer cells in each section was measured using ImageJ 1.45. For each section, the quantification of cleaved-caspase 3-positive cells was determined by counting the number of tumor cells in five different fields of each section and referred to the total number of counted cancer cells.

For immunohistochemistry staining, the following antibodies were used: anti-SENP1 (Abcam, ab108981, 1:200), anti-cleaved-caspase 3 (Cell Signaling Technology, 9661, 1:100), anti-hexokinase 2 (Proteintech, 22029-1-AP, 1:200).

**Statistics and reproducibility.** The cell data are presented as mean ± SEM (standard error of the mean) of triplicate wells from one representative experiment. All immunoblots from cell samples were repeated at least three times independently with similar results. Comparisons between the two groups were performed by unpaired two-tailed Student's $t$ test. Correlations between groups were determined by the chi-square test. Survival rates were analyzed by the Kaplan–Meier method. The sample number ($n$) indicates the number of independent biological samples in each experiment. Generally, all experiments were carried out with $n \geq 3$ biological replicates. Analyses were performed using GraphPad Prism 8.0 software. *$P \leq 0.05$; **$P \leq 0.01$; ***$P \leq 0.001$; NS, not significant.

**Study approval.** The use of pathological specimens and the review of all pertinent patient records were approved by the Shanghai Jiao Tong University School of Medicine, Renji Hospital Ethics Committee (RA-2019-241). All participants provided written informed consent for the use of their samples for research purposes. Our study is compliant with the "Guidance of the Ministry of Science and Technology (MOST) for the Review and Approval of Human Genetic Resources", which requires formal approval for the export of human genetic material or data from China. All animal experiments were performed in compliance with the Guide for the Care and Use of Laboratory Animals (National Academies Press, 2011) and were approved by the Animal Care Committee of Shanghai Jiao Tong University School of Medicine. The animal study was also performed according to the ARRIVE guidelines.

**Reporting summary.** Further information on research design is available in the Nature Research Reporting Summary linked to this article.

## Data availability

The gene expression profile of GSE83654, GSE33455, and GSE36135 has obtained from the National Center for Biotechnology Information (NCBI) Genome database (https://www.ncbi.nlm.nih.gov). The Cancer Genome Atlas human prostate cancer data set (TCGA Firehose Legacy) was downloaded from cBioPortal for Cancer Genomics (http://cbioportal.org). All other data supporting the findings of this study are available from the corresponding author upon reasonable request. A Reporting Summary for this study is available as a Supplementary Information file. Source data are provided with this paper.

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

## Acknowledgements

We sincerely appreciate Prof. Nissim Hay for sharing plenti6-rat*HK2*-HA-WT, rat*HK2*-HA-MTD, and rat*HK2*-HA-SA plasmids. The work was supported by the National Natural Science Foundation of China No. 81702542 (Q.W.), No. 81972578 (Q.W.), 81703533 (W.Z.), and 82072847 (W.X.); the Program for Professor of Special Appointment (Eastern Scholar) at Shanghai Institutions of Higher Learning No. TP2017029 (Q.W.); Shanghai Municipal Education Commission-Gaofeng Clinical Medicine Grant Support No. 20171912 (Q.W.). The funders had no role in study design, data collection and analysis, decision to publish, or preparation of the paper.

## Author contributions

Q.W., J.C., and W.X. conceived and designed the experiments. X.S. and J.H. performed most of the experiments and analyzed the data; Z.M., W.Z., Y.J., K.S., Z.Y., W.L., Z.X., Q. Z., Y.C., J.P., and B.D. performed a specific subset of the experiments and analyses; Q.W. and X.S. wrote the paper.

## Competing interests

The authors declare no competing interests.
