## [Peer Review File · Nature Communications]

Reviewers' comments:

Reviewer #1 (Remarks to the Author); expert in metabolism, hexokinases, mitochondria:

In this manuscript the authors found that hexokinase 2 (HK2) is sumoylated. The sumoylation appears to decrease binding to mitochondria. A mutant of HK2 resistant to sumoylation binds more avidly to mitochondria with concomitant increase in glycolysis. This HK2 mutant also increased prostate cancer cell proliferation and resistance to chemotherapy. Overall the studies are thorough and convincing. I have only minor points :

1. The effect of HK2 on ROS is not necessarily due to the effect on OXPHO but could also be through an increase flux to the PPP.
2. Hk2 binds to the OMM and interacts with VDAC. This was not illustrated correctly in the graphic abstract.
3. At would be nice to know at what conditions HK2 is sumoylated or de-sumoyl

Reviewer #2 (Remarks to the Author); expert in SUMO:

Major comments

1 - I am puzzled by the idea of the authors that Hexokinase2 contains two sumoylation sites (lysines 315 and 492) that can be mutated individually without affecting its sumoylation, but when both mutations are combined, sumoylation of the protein is lost (Figure 1E). If both lysines can act as SUMO acceptor sites, it should be possible to observe a doubly modified form of Hexokinase2. However, the authors never identify more than one SUMO-modified band in their experiments. Their finding that lysines 315 and 492 are conserved amongst different species (Figure 1D) could indicate that these lysines are structurally important. When mutating both lysines, the protein could lose its sumoylation because of a loss of structure. The double mutant should therefore be rigorously validated to show convincingly that its structure is not affected.

In search of alternative evidence for sumoylation of Hexokinase2, I found Hexokinase2 in one of the major site-specific SUMO proteomics screen (Hendriks et al. 2017 Nature Struct Mol Biol). The identified sumoylation site of Hexokinase2 in this paper is lysine 147 in the motif DKK(sumo)LPL, which matches the inverted sumoylation motif [ED]xK[psi]. Hendriks et al. found that the sumoylation of Hexokinase2 is increased upon blocking the proteasome. This result is inconsistent with the results of the authors who propose lysines 315 and 492 as major SUMO acceptor sites. The authors should verify whether the K147R mutant of Hexokinase2 is reduced for sumoylation and carry out their functional experiments with the K147R mutant instead of the K315R K492R double mutant. The authors should also test a E317A E494A double mutant of Hexokinase2.

2 - The IP Westerns used by the authors to demonstrate sumoylation of Hexokinase2 are not convincing. When performing IPs, a pitfall is the presence of cysteine-bridged heavy chains that run at similar height as the band claimed by the authors to be sumoylated Hexokinase2. His-tagged SUMO can efficiently be enriched under denaturing conditions and SUMO-modified proteins can subsequently be detected efficiently by immunoblotting (Tatham et al. 2009 Nature Protocols) without any of the antibody problems pointed out.

3 - For the IP Westerns demonstrating sumoylation, authors should include full size blots for IPs and inputs of the Hexokinase2 to allow an estimate of the sumoylation efficiency. Moreover, equal amounts of antibody should be used for the specific IP and the control IP. Some of the figures, e.g. 1C and 1F show lower amounts of antibody for the control IP.

Minor comments

4 - The paper would benefit from correction by a native speaker since it contains frequent style

and language errors.

5 - SENP2 knockdown in Figure 1G is not convincing, since one of the controls shows a similar reduction.

Reviewer #3 (Remarks to the Author): expert in prostate cancer mouse models, signaling, metabolism:

This manuscript examines the regulation of HK2 association with mitochondria as regulated by sumoylation, the effect of sumoylation defective HK2 on glycolysis, and the consequences of decreased HK2 sumoylation for response to docetaxel. The authors demonstrate that sumoylation occurs on K315 or K492 of HK2 and that loss of both sumoylation sites leads to reduced VDAC1 association on mitochondria. Over-expression and depletion studies were used to nominate SENP1 as the dominant de-sumoylation enzyme. In AR negative PC3 prostate cancer cells, loss of HK2 sumoylation and/or mitochondrial association was correlated with increased glycolysis, decreased oxygen consumption, a faster growth rate in vitro and in vivo, and decreased response to docetaxel in vitro. SENP1 demonstrated increased expression following docetaxel treatment of PC3 cells. In addition high HK2 and SENP1 expression was associated with higher Gleason grade primary prostate cancer patient samples, and there was a small statistical correlation with high HK2/SENP1 expression in post-treatment samples and shorter biochemical recurrence response to neoadjuvant chemo-ADT.

This manuscript contributes to an increased understanding of the biochemical regulation of HK2, an important enzyme in cancer growth. It also presents an internally-consistent picture of HK2 associated biology in PC3 cells. PC3 cells are AR negative, poorly differentiated cells that are a wholly inaccurate model of primary prostate cancer, but instead, may represent a relatively small subclass of castration resistant prostate cancer. The extension of the PC3 data to the interpretation of primary prostate cancer metabolic vulnerabilities and drug sensitivity is not appropriate as outlined below.

1. Although the effects of docetaxel treatment in vitro on genetically modified PC3 cells was shown, in vivo responses to docetaxel were not shown- this is an omission. Other than utilizing mouse organoid models, it is quite difficult to model primary human prostate cancer. However, analyzing the effect of genetically modified HK2 expression in LNCaP and determining the in vivo response to ADT, docetaxel, and ADT+docetaxel would be much more comparable to primary neoadjuvant data.
2. Increased HK2 and SENP1 expression is associated with higher Gleason grades in primary prostate cancer. Therefore, it is predicted that there will be poorer biochemical progression free survival (BPFS) in this subclass. This is true for a large number of tissue markers that increase with Gleason grade. One approach with sufficiently high sample numbers (such as exists for TCGA) is to analyze Gleason grade 7 only with respect to HK2, SENP1, and BPFS. Alternatively, regression analyses to establish the contribution of independent variables could be done. The analysis as presented in Figs. 6 and S6 is not particularly useful.
3. For the neoadjuvant chemo-ADT therapy analysis shown in Figure 6D, the numbers of samples are relatively small and the actual numbers should be shown as opposed to percentage. Because the SENP1 and HK2 levels were determined after treatment, these markers are not measuring susceptibility.
4. Additional comments:
 - a. The suggestion that sumoylation occurs at one of two alternative sites, that the sites are in distinct regions of the protein, and that sumoylation of either site alone appears to be sufficient for function is not addressed.
 - b. Figures S3 E&F need negative controls (T473A and DKR, respectively) due to the potential for high non-specific binding.

First, we would like to thank all the reviewers for taking the time to review our manuscript. Please find below a point-to-point response to the reviewers' comments. For convenience, we have repeated the relevant new data for each reviewer.

Reviewers' comments:

Reviewer #1 (Remarks to the Author); expert in metabolism, hexokinases, mitochondria:

In this manuscript the authors found that hexokinase 2 (HK2) is sumoylated. The sumoylation appears to decrease binding to mitochondria. A mutant of HK2 resistant to sumoylation binds more avidly to mitochondria with concomitant increase in glycolysis. This HK2 mutant also increased prostate cancer cell proliferation and resistance to chemotherapy. Overall the studies are thorough and convincing. I have only minor points:

1. The effect of HK2 on ROS is not necessarily due to the effect on OXPHO but could also be through an increase flux to the PPP.

In reply: Thanks for pointing out that the effect of HK2 on ROS. As we mentioned in the instruction, the product of HK2 kinase activity, glucose-6-phosphate, can be used for PPP and reduce cellular ROS. Our intention was to state that binding of HK2 to mitochondria reduce global cellular ROS and ROS from mitochondria. We have changed the sentence to "While MTD HK2 moderately increased intracellular ROS level, partially from mitochondria, K315/492R HK2 decreased both intracellular and mitochondrial ROS levels".

2. Hk2 binds to the OMM and interacts with VDAC. This was not illustrated correctly in the graphic abstract.

In reply: We thank the reviewer to highlight this oversight. We have corrected the graph accordingly.

3. At would be nice to know at what conditions HK2 is sumoylated or de-sumoyl

In reply: The progress of SUMOylation is generally dynamic and reversible. In most cases, SENPs have a critical role in maintaining a balance between the level of SUMOylation. In our results, docetaxel treatment increases cellular ROS level, leads to active HIF1 pathway, and induces SENP1 expression. SENP1 subsequently deSUMOylates HK2 and other target proteins. Therefore, the HK2 deSUMOylation may be a response for cancer cell under chemo-therapy.

Reviewer #2 (Remarks to the Author); expert in SUMO:

Major comments

1 - I am puzzled by the idea of the authors that Hexokinase2 contains two sumoylation sites (lysines 315 and 492) that can be mutated individually without affecting its sumoylation, but when both mutations are combined, sumoylation of the protein is lost (Figure 1E). If both lysines can act as SUMO acceptor sites, it should be possible to observe a doubly modified form of Hexokinase2. However, the authors never identify more than one SUMO-modified band in their experiments. Their finding that lysines 315 and 492 are conserved amongst different species (Figure 1D) could indicate that these lysines are structurally important. When mutating both lysines, the protein could lose its sumoylation because of a loss of structure. The double mutant should therefore be rigorously validated to show convincingly that its structure is not affected. In search of alternative evidence for sumoylation of Hexokinase2, I found Hexokinase2 in one of the major site-specific SUMO proteomics screen (Hendriks et al. 2017 Nature Struct Mol Biol). The identified sumoylation site of Hexokinase2 in this paper is lysine 147 in the motif DKK(sumo)LPL, which matches the inverted sumoylation motif [ED]xK[psi]. Hendriks et al. found that the sumoylation of Hexokinase2 is increased upon blocking the proteasome. This result is inconsistent with the results of the authors who propose lysines 315 and 492 as major SUMO acceptor sites. The authors should verify whether the K147R mutant of Hexokinase2 is reduced for sumoylation and carry out their functional experiments with the K147R mutant instead of the K315R K492R double mutant. The authors should also test a E317A E494A double mutant of Hexokinase2.

In reply: We appreciate the comments. We understand that there is a possibility that both sites can be SUMOylated. However, our results showed only one molecule of SUMO conjugated to HK2, either K315 or K492. These two lysines are not simultaneously SUMOylated in cells. One of the possible explanations is, one molecule SUMO1 is enough to prevent HK2 interacting with VDAC1. We list some articles from different labs reported similar cases as following. As a 101-amino acids protein, SUMO1 conjugates to target only one molecule even other available sites exist.

(1-5)

1. Li J, Lu D, Dou H, Liu H, Weaver K, Wang W, et al. Desumoylase SENP6 maintains osteochondroprogenitor homeostasis by suppressing the p53 pathway. *Nature Communications*. 2018;9.
2. Qiu C, Wang Y, Zhao H, Qin L, Shi Y, Zhu X, et al. The critical role of SENP1-mediated GATA2 deSUMOylation in promoting endothelial activation in graft arteriosclerosis. *Nature Communications*. 2017;8.
3. Huang J, Yan J, Zhang J, Zhu SG, Wang YL, Shi T, et al. SUMO1 modification of PTEN regulates tumorigenesis by controlling its association with the plasma membrane. *Nature Communications*. 2012;3.
4. Zhu X, Ding S, Qiu C, Shi Y, Song L, Wang Y, et al. SUMOylation Negatively Regulates Angiogenesis by Targeting Endothelial NOTCH Signaling. *Circulation Research*. 2017;121(6):636-+.
5. Bao J, Qin M, Mahaman YAR, Zhang B, Huang F, Zeng K, et al. BACE1 SUMOylation increases its stability and escalates the protease activity in Alzheimer's disease. *Proceedings of the National*

Again, thanks for pointing out that mutant of K315/492R may affect the SUMOylation because of loss of structure. We generated human HK2 3D protein models from SWISS-MODEL (<https://swissmodel.expasy.org/>). These models show no structure difference between wild-type and K315/492R mutant. In addition, we generated E317/494A mutant (DEA), and found that SUMOylation of HK2 reduced notably, compared to those of wild-type (Figure S2D). These data, combined with our previous data, strongly suggest that K315 and K492 is two SUMOylated sites for HK2.

K147 is a reversed potential SUMO site in HK2. With GPS-SUMO analysis (<http://sumosp.biocuckoo.org/online.php>), the score of K147 is much lower than K315 and K492. As requested, we also generated K147R mutant HK2 and tested by the same His-SUMO IP protocol. However, K147R HK2 did not reduce SUMO band compared with that of wild-type. These data suggest K147 may not be available for SUMOylation.

Position	Peptide	Score	P-Value
147	KLQIKDKLPLGFTF	1.36	0.91
315	LILVKMAKEELLFGG	3.85	0.035
492	LEVKRRMKVEMERGL	13.1	0.007

2 - The IP Westerns used by the authors to demonstrate sumoylation of Hexokinase2 are not convincing. When performing IPs, a pitfall is the presence of cysteine-bridged heavy chains that run at similar height as the band claimed by the authors to be sumoylated Hexokinase2. His-tagged SUMO can efficiently be enriched under denaturing conditions and SUMO-modified proteins can subsequently be detected efficiently by immunoblotting (Tatham et al. 2009 Nature Protocols) without any of the antibody problems pointed out.

In reply: New IP data using His-tagged SUMO have been added as requested. We have now included these data in Figure1. The results agree with our previous observation with HA-tagged SUMO and endogenous SUMOylation of HK2.

3 - For the IP Westerns demonstrating sumoylation, authors should include full size blots for IPs and inputs of the Hexokinase2 to allow an estimate of the sumoylation efficiency. Moreover, equal amounts of antibody should be used for the specific IP and the control IP. Some of the figures, e.g. 1C and 1F show lower amounts of antibody for the control IP.

In reply: We included all the full-size Western blotting in the source data. We agree with the reviewer that the antibody amount in specific IP is not ideal. We attempt to use same amount of

antibody and IgG for IP, but hampered by the technical limitation.

Minor comments

4 - The paper would benefit from correction by a native speaker since it contains frequent style and language errors.

In reply: We thank the reviewer for this suggestion. We have had the professional scientific editing team help us revise this manuscript. We are more than happy to further edit the text as the reviewers and/or editor see fit.

5 - SENP2 knockdown in Figure 1G is not convincing, since one of the controls shows a similar reduction.

In reply: We have performed all the Western Blotting about SENP2 and include these data in Figure1G. The results show that SENP2 reduces only in knockdown cells.

Reviewer #3 (Remarks to the Author): expert in prostate cancer mouse models, signaling, metabolism:

This manuscript examines the regulation of HK2 association with mitochondria as regulated by sumoylation, the effect of sumoylation defective HK2 on glycolysis, and the consequences of decreased HK2 sumoylation for response to docetaxel. The authors demonstrate that sumoylation occurs on K315 or K492 of HK2 and that loss of both sumoylation sites leads to reduced VDAC1 association on mitochondria. Over-expression and depletion studies were used to nominate SENP1 as the dominant de-sumoylation enzyme. In AR negative PC3 prostate cancer cells, loss of HK2 sumoylation and/or mitochondrial association was correlated with increased glycolysis, decreased oxygen consumption, a faster growth rate in vitro and in vivo, and decreased response to docetaxel in vitro. SENP1 demonstrated increased expression following docetaxel treatment of PC3 cells. In addition high HK2 and SENP1 expression was associated with

higher Gleason grade primary prostate cancer patient samples, and there was a small statistical correlation with high HK2/SEN1 expression in post-treatment samples and shorter biochemical recurrence response to neoadjuvant chemo-ADT.

This manuscript contributes to an increased understanding of the biochemical regulation of HK2, an important enzyme in cancer growth. It also presents an internally-consistent picture of HK2 associated biology in PC3 cells. PC3 cells are AR negative, poorly differentiated cells that are a wholly inaccurate model of primary prostate cancer, but instead, may represent a relatively small subclass of castration resistant prostate cancer. The extension of the PC3 data to the interpretation of primary prostate cancer metabolic vulnerabilities and drug sensitivity is not appropriate as outlined below.

1. Although the effects of docetaxel treatment in vitro on genetically modified PC3 cells was shown, in vivo responses to docetaxel were not shown- this is an omission. Other than utilizing mouse organoid models, it is quite difficult to model primary human prostate cancer. However, analyzing the effect of genetically modified HK2 expression in LNCaP and determining the in vivo response to ADT, docetaxel, and ADT+docetaxel would be much more comparable to primary neoadjuvant data.

In reply: We thank the reviewer for pointing out that omission. We agree with this reviewer that using AR sensitive cell lines for mouse model with ADT+docetaxel treatment would be more comparable. We attempted to conduct the in vivo experiment using androgen sensitive cells with different mutant HK2. Unfortunately, the animal facility is locked down because of the COVID-19 pandemic and we are unable to conduct the suggested experiment. However, as shown in Figure S5E, we LNCaP cells in charcoal-stripped medium (mimicking androgen deprivation) with or without docetaxel treatment (Figure S5E). This indirect evidence supported our previous conclusions that SUMO-defective HK2 protects prostate cancer cells from docetaxel stress.

2. Increased HK2 and SENP1 expression is associated with higher Gleason grades in primary prostate cancer. Therefore, it is predicted that there will be poorer biochemical progression free survival (BPFS) in this subclass. This is true for a large number of tissue markers that increase with Gleason grade. One approach with sufficiently high sample numbers (such as exists for TCGA) is to analyze Gleason grade 7 only with respect to HK2, SENP1, and BPFS. Alternatively, regression analyses to establish the contribution of independent variables could be done. The analysis as presented in Figs. 6 and S6 is not particularly useful.

In reply: Based on the reviewer's suggestion, we analyzed the subgroup with Gleason score=7 with

TCGA data (Figure S6C). Consistent with our tissue microarray, high HK2 and SENP2 level also indicated the poor prognosis.

3. For the neoadjuvant chemo-ADT therapy analysis shown in Figure 6D, the numbers of samples are relatively small and the actual numbers should be shown as opposed to percentage. Because the SENP1 and HK2 levels were determined after treatment, these markers are not measuring susceptibility.

In reply: The actual patient number was added to Figure 6D. We agree with the reviewer’s point that the numbers are relatively small. Neoadjuvant chemo-ADT therapy is currently being conducted in our department and we will analyze these data in the future.

4. Additional comments:

a. The suggestion that sumoylation occurs at one of two alternative sites, that the sites are in distinct regions of the protein, and that sumoylation of either site alone appears to be sufficient for function is not addressed.

In reply: We appreciate the comments. We understand that there is a possibility that both sites can be SUMOylated. However, our results showed only one molecule of SUMO conjugated to HK2, either K315 or K492. These two lysines are not simultaneously SUMOylated in cells. One of the possible explanations is, one molecule SUMO1 is enough to prevent HK2 interacting with VDAC1. We list some articles from different labs reported similar cases as following. As a 101-amino acids protein, SUMO1 conjugates to target only one molecule even other available sites exist.

(1-5)

1. Li J, Lu D, Dou H, Liu H, Weaver K, Wang W, et al. Desumoylase SENP6 maintains osteochondroprogenitor homeostasis by suppressing the p53 pathway. *Nature Communications*. 2018;9.
2. Qiu C, Wang Y, Zhao H, Qin L, Shi Y, Zhu X, et al. The critical role of SENP1-mediated GATA2 deSUMOylation in promoting endothelial activation in graft arteriosclerosis. *Nature Communications*. 2017;8.
3. Huang J, Yan J, Zhang J, Zhu SG, Wang YL, Shi T, et al. SUMO1 modification of PTEN regulates tumorigenesis by controlling its association with the plasma membrane. *Nature Communications*. 2012;3.
4. Zhu X, Ding S, Qiu C, Shi Y, Song L, Wang Y, et al. SUMOylation Negatively Regulates Angiogenesis by Targeting Endothelial NOTCH Signaling. *Circulation Research*. 2017;121(6):636-+.
5. Bao J, Qin M, Mahaman YAR, Zhang B, Huang F, Zeng K, et al. BACE1 SUMOylation increases its stability and escalates the protease activity in Alzheimer's disease. *Proceedings of the National Academy of Sciences of the United States of America*. 2018;115(15):3954-9.

b. Figures S3 E&F need negative controls (T473A and DKR, respectively) due to the potential for high non-specific binding.

In reply: We added T473A and DKR to Figure S3 as requested.

REVIEWER COMMENTS

Reviewer #1 (Remarks to the Author):

The authors adequately addressed my concerns.

Reviewer #2 (Remarks to the Author):

The authors have properly addressed my concerns.

Reviewer #3 (Remarks to the Author):

This paper contributes to a deeper understanding of the biochemical regulation of HK2 and related consequences for in vitro metabolism. The suggestion that the regulation of HK2 sumoylation may play a role in prostate cancer response to docetaxel is not supported by either the model system or the correlative clinical data presented here. Given that it has not been possible to perform in vivo studies, the conclusions have rested on in vitro studies and a preliminary neoadjuvant clinical study, which are discussed further below.

As outlined in the initial review of this paper, PC3 cells as a model system do not have a clinical correlate except possibly as a double negative highly plastic dedifferentiation model. This is a rare class of castrate resistant prostate cancer and does not represent a model for the primary prostate cancer analysis in Figure 6. Nevertheless, these cells have been useful for the metabolic studies. However, the in vitro docetaxel response (< 50% killing at 20 nM, Fig 5G & H) and the change relative to the expression of HK2 mutants was not convincing. Unfortunately, it has not been possible to perform in vivo studies, which are necessary at this point given the less than convincing in vitro response. A statistically significant change (in Fig 5H, ~10% to ~7% death) is not clinically meaningful.

In response to the first review, the authors have provided in vitro growth response data with an additional model, WT and HK2 mutant LNCaP cells treated with ADT and docetaxel. These cells are closer in phenotype to the presented clinical data. Unfortunately, there was no further analysis in LNCaP cells of their hypothesis that HK2 sumoylation and HIF-initiated SENP1 induction were related to survival following DXT and DXT/ADT treatment.

Also as stated earlier, high HK2 and SENP1 expression appear to be correlated with high Gleason grade and therefore, the association of HK2/SENP1 expression with biochemical recurrence is expected. There are tens (if not hundreds) of immunohistochemical/RNA markers that increase with Gleason grade. HK2 and SENP1 are likely not significant as potentially important clinical markers, but the information may contribute to future interpretations of physiology relative to Gleason grade.

Finally, the authors have provided numbers of patients for the preliminary clinical trial analyzing neoadjuvant ADT/DXT response relative to HK2/SENP1 expression (Figure 6D). This is a trial of previously untreated primary prostate cancer. The authors attempt to provide a clinical study is laudatory. However, I don't feel that the results of the study support their conclusions about the translational significance of the present data. Figure 6D is slightly different as shown in the response letter to the reviewers compared to the manuscript. In the former, the inclusion of a few more patients resulted in a loss of any statistical significance. Although I am not clear about which is the actual figure, the data is too preliminary to appropriately include.

Reviewer Comments

Reviewer #3 (Remarks to the Author):

This paper contributes to a deeper understanding of the biochemical regulation of HK2 and related consequences for *in vitro* metabolism. The suggestion that the regulation of HK2 sumoylation may play a role in prostate cancer response to docetaxel is not supported by either the model system or the correlative clinical data presented here. Given that it has not been possible to perform *in vivo* studies, the conclusions have rested on *in vitro* studies and a preliminary neoadjuvant clinical study, which are discussed further below.

We sincerely thank reviewer #3 for the detailed and careful reviews. You provided a number of constructive and insightful comments to improve our manuscript. We have made major changes to the manuscript including the execution of a series of *in vivo* experiments and associated analysis. Please find below a point-by-point response to the reviewer's comments.

As outlined in the initial review of this paper, PC3 cells as a model system do not have a clinical correlate except possibly as a double negative highly plastic dedifferentiation model. This is a rare class of castrate resistant prostate cancer and does not represent a model for the primary prostate cancer analysis in Figure 6. Nevertheless, these cells have been useful for the metabolic studies. However, the *in vitro* docetaxel response (< 50% killing at 20 nM, Fig 5G & H) and the change relative to the expression of HK2 mutants was not convincing. Unfortunately, it has not been possible to perform *in vivo* studies, which are necessary at this point given the less than convincing *in vitro* response. A statistically significant change (in Fig 5H, ~10% to ~7% death) is not clinically meaningful.

First, we have performed the IC50 assay to decide the docetaxel concentration in PC3 and finally chose 20nM in our experiment. *In vitro*, docetaxel functions to arrest cell proliferation rather than cell apoptosis or death. That's why in Figure S5E, even for control cells (scramble shRNA), only 10% suffered apoptosis or death. As we mentioned in the discussion, considering the high level of SENP1 in PC3 cells, HK2 prone to be de-SUMOylated and binds to the mitochondria. So, the difference between wild-type HK2 and SUMO-defective HK2 may not be so impressive. It is worth noting that with docetaxel treatment, cells suffered more apoptosis or death when HK2 detached from mitochondria (HK2-MTD, 15%), compared with HK2 binding to mitochondria (HK2-DKR, 7%), almost doubling level. Moreover, we developed PC3-derived xenografts in nude mice with or without docetaxel treatment and showed the SUMO-defective HK2 conferred a survival advantage compared with HK2 wild-type. As requested by the reviewer, for hormone-sensitive prostate cancer model, our initial strategy to answer this question was to inject LNCaP cells with different HK2 mutant forms to nude mice subcutaneously. However, although we were able to successfully establish the cell lines and pursue the *in vitro* experiments with docetaxel and/or ADT treatment (Figure 5H), we found it difficult to develop xenografts in the nude mice given the poor tumorigenicity of LNCaP cells. Alternatively, we chose another PSA positive cell lines 22Rv1 and showed that tumors with SUMO-defective HK2 offered a survival advantage and eliminated cell apoptosis with docetaxel treatment, in intact or castrated.

Figure S5E

Figure S5G and H

Figure S5I and J

In response to the first review, the authors have provided in vitro growth response data with an additional model, WT and HK2 mutant LNCaP cells treated with ADT and docetaxel. These cells are closer in phenotype to the presented clinical data. Unfortunately, there was no further analysis in LNCaP cells of their hypothesis that HK2 sumoylation and HIF-initiated SENP1 induction were related to survival following DXT and DXT/ADT treatment.

We performed Western blotting to detect the HIF1 α and SENP1 expression in LNCaP cells under docetaxel treatment, with or without ADT. These results, shown in Figure S5C, are consistent with our conclusion that docetaxel upregulates HIF1 α and SENP1 expression, which promotes HK2 binding to mitochondria. In addition, we also used 22Rv1 cells to perform the Western blotting and the results were consistent.

Also as stated earlier, high HK2 and SENP1 expression appear to be correlated with high Gleason grade and therefore, the association of HK2/SENP1 expression with biochemical recurrence is expected. There are tens (if not hundreds) of immunohistochemical/RNA markers that increase with Gleason grade. HK2 and SENP1 are likely not significant as potentially important clinical markers, but the information may contribute to future interpretations of physiology relative to Gleason grade.

We agree with the reviewer that high HK2 and SENP1 expression is associated with tumor aggressiveness. Actually, we do not attempt to consider HK2 and SENP1 as clinical markers because of the lack of approach to detecting HK2 SUMOylation status conveniently. Instead, we interest in the mechanism of HK2 in tumorigenesis and therapy progression. In this study, we focus on the relation between HK2 SUMOylation and subcellular localization, and the role of this interaction in tumorigenesis and therapy response. We will explore the deeper mechanism of HK2 in cell behavior, especially crosstalk with tumor environment. Thank you again for your constructive comments.

Finally, the authors have provided numbers of patients for the preliminary clinical trial analyzing neoadjuvant ADT/DXT response relative to HK2/SENP1 expression (Figure 6D). This is a trial of previously untreated primary prostate cancer. The authors attempt to provide a clinical study is laudatory. However, I don't feel that the results of the study support their conclusions about the translational significance of the present data. Figure 6D is slightly different as shown in the response letter to the reviewers compared to the manuscript. In the former, the inclusion of a few more patients resulted in a loss of any statistical significance. Although I am not clear about which is the actual figure, the data is too preliminary to appropriately include.

Thank you for your careful reading. We apologize for our error. We used two cohorts of tissue microarray in this study. One for Figure 6B and 6C, including 121 patients without any neoadjuvant therapy. The clinical information was shown in Table S4. The other cohort includes 116 prostate cancer patients with neoadjuvant therapy and shown in Figure 6D and S6D. When doublechecked the patient information in Figure 6D, we found there were 2 patients have double cores, and another 6 cores are from metastatic lymph nodes. We removed these 8 cores from the analysis have made the changes in the finalized manuscript. The clinical information was shown in Table S5. We apologize for not addressing clearly the difference in the previous revised response letter and thank you again for your kind reminder.

REVIEWERS' COMMENTS

Reviewer #3 (Remarks to the Author):

As discussed previously, this manuscript makes interesting observations with respect to the biochemical regulation of HK2. Based on tumor response characteristics, the in vivo cell line xenograft data appears to be most consistent with the conclusion that sumo defective HK2 promotes growth but has a relatively neutral effect on docetaxel treatment response. The cleaved caspase 3 density is difficult to assess- I was not able to find a description in the methods. An assay for relative BrDU incorporation would be cleaner. As a note, 22RV1 is not comparable to the clinical data, despite being AR and PSA positive, because this cell line is castration resistant and primary prostate cancer is castration sensitive. It doesn't really add much. Finally, I would suggest that "Moreover, we highlighted that low HK2 expression increases the sensitivity of prostate cancer patients to docetaxel treatment and that activation of the SENP1-HK2 axis desensitizes chemotherapy response in prostate cancer cells" be restated to summarize a demonstration of correlation.

REVIEWERS' COMMENTS

Reviewer #3 (Remarks to the Author):

As discussed previously, this manuscript makes interesting observations with respect to the biochemical regulation of HK2. Based on tumor response characteristics, the in vivo cell line xenograft data appears to be most consistent with the conclusion that sumo defective HK2 promotes growth but has a relatively neutral effect on docetaxel treatment response. The cleaved caspase 3 density is difficult to assess- I was not able to find a description in the methods. An assay for relative BrDU incorporation would be cleaner. As a note, 22RV1 is not comparable to the clinical data, despite being AR and PSA positive, because this cell line is castration resistant and primary prostate cancer is castration sensitive. It doesn't really add much. Finally, I would suggest that "Moreover, we highlighted that low HK2 expression increases the sensitivity of prostate cancer patients to docetaxel treatment and that activation of the SENP1-HK2 axis desensitizes chemotherapy response in prostate cancer cells" be restated to summarize a demonstration of correlation.

We would like to thank the reviewer3 for taking the time to review for a third time our manuscript. To show role of SUMO defective HK2 in response to docetaxel treatment, we have added in vivo experiments as suggested by the reviewer. We admit that 22Rv1 is not the perfect cell line to mimic the castration sensitive prostate cancer because of the alternative androgen receptor pathway activation. We attempt to use LNCaP cells in the xenograft tumor experiment. Unfortunately, LNCaP cells in our lab is not aggressive enough to form xenograft. Our plan B is to do the in vitro experiments with LNCaP, 22Rv1, and PC3, and in vivo experiments with 22Rv1 and PC3. Indeed, 22Rv1 are not completely castration resistance. We have still observed the response to castration in 22Rv1, as shown in Figure S5i.

For Cleaved-Caspase 3 quantification, we collected all the tumors from mice and stained with Cleaved-Caspase 3 antibody. Five non-overlapping fields in each section of mice tumor were analyzed in blinded manner. The integral optical density of immunopositive cancer cells in each section was measured using imageJ. For each section, quantification of Cleaved-Caspase 3 positive cells was determined by counting the number of tumors cells in five different fields of each section, and referred to the total number of counted cancer cells.

Finally, we agree with the reviewer. As suggested, we have toned down the writing. The revised manuscript reads as follows: "Moreover, we demonstrate an inverse relationship between SENP1-hexokinase 2 axis and chemotherapy response in prostate cancer samples."

Thank you again for your effort and very helpful comments, which have helped us to improve our paper.